# WASA: WAtermark-based Source Attribution for Large Language Model-Generated Data

## Abstract

The impressive performances of *large language models* (LLMs) and their immense potential for commercialization have given rise to serious concerns over the *intellectual property* (IP) of their training data. In particular, the synthetic texts generated by LLMs may infringe the IP of the data being used to train the LLMs. To this end, it is imperative to be able to (a) identify the data provider who contributed to the generation of a synthetic text by an LLM (**source attribution**) and (b) verify whether the text data from a data provider has been used to train an LLM (**data provenance**). In this paper, we show that both problems can be solved by watermarking, i.e., by enabling an LLM to generate synthetic texts with embedded watermarks that contain information about their source(s). We identify the key properties of such watermarking frameworks (e.g., source attribution accuracy, robustness against adversaries), and propose a *WAtermarking for Source Attribution* (WASA) framework that satisfies these key properties due to our algorithmic designs. Our WASA framework enables an LLM to learn an accurate mapping from the texts of different data providers to their corresponding unique watermarks, which sets the foundation for effective source attribution (and hence data provenance). Extensive empirical evaluations show that our WASA framework achieves effective source attribution and data provenance.

## 1 Introduction

*Large language models* (LLMs) (Ouyang et al., 2022; Touvron et al., 2023) have recently demonstrated remarkable performances and hence received a surging interest. These LLMs, which are trained using massive text data, have displayed impressive text generation abilities. This has given rise to the immense potential of adopting the texts generated by LLMs for commercial use. However, this potential commercialization has led to major concerns regarding the *intellectual property* (IP) of training data for LLMs because the texts generated by an LLM may infringe the IP of the data being used to train the LLM. These concerns have been reflected by the increasing regulations on data protection related to AI models. For example, the Coalition for Content Provenance and Authenticity has stressed the necessity of certifying the *source* of online content produced by generative AI models (Rosenthol, 2022). Therefore, it is of crucial importance for LLMs to be equipped with **source attribution** and **data provenance** for their generated synthetic texts.

In **source attribution**, given some synthetic texts generated by an LLM, its aim is to find the source responsible for the generation of these texts. That is, if the data from a data provider has been used to train the LLM and contributed to the generation of a sentence by the LLM, then source attribution identifies this data provider. Moreover, source attribution also improves the interpretability of LLM-generated texts: For example, if the generated content from an LLM is attributed to a trustworthy source (e.g., a peer-reviewed academic paper), then the user is likely to consider the content more reliable. The ability to perform source attribution also endows the LLM with the capability of **data provenance**. Specifically, a data provider can check the source of the generated texts from an LLM via source attribution, and hence verify whether its data has been used to train the LLM, as detailed in Sec. 3.4. This has important implications for data protection. As an example, patients who refuse to have their medical data used for training an LLM for medical diagnosis (due to privacy concerns) may request their data provenance to check whether their data has been misused.

To perform source attribution (and hence achieve data provenance) for LLM-generated synthetic texts, a natural solution involves *watermarking*, i.e., by enabling the LLM to generate synthetic texts

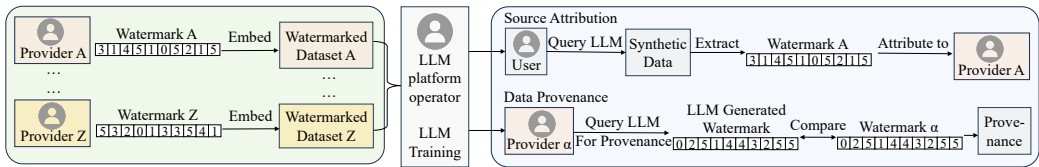

Figure 1: Illustration of `WASA`'s problem setting. Watermarks are embedded into the texts from data providers for training the LLM. The LLM produced by our `WASA` framework can generate synthetic texts with embedded watermarks that allow for effective source attribution and data provenance.

with embedded watermarks that contain information about their source(s). Consequently, source attribution can be performed by examining the watermarks embedded into the generated texts. Our problem setting (Fig. 1) involves 3 parties: *data providers* contributing text data that may be used for LLM training, an honest third-party *LLM platform operator* producing an LLM with generated texts that embed watermarks (hence allowing for source attribution), and *users* of the texts generated by this LLM. The users may request **source attribution** for the LLM-generated synthetic texts to find out which data provider is responsible for the generated texts, while the data providers may request **data provenance** to verify whether their contributed data has been used to train the LLM. We consider scenarios where each data provider contributes ample balanced data with unique characteristics, i.e., the data from different data providers exhibit dissimilarities. This encompasses a wide variety of real-world scenarios: For example, online articles written by different authors (i.e., data providers) usually feature their unique writing styles. On the other hand, we do not consider individual documents/sentences as data providers in our setting since they have insufficient data.

A watermarking framework capable of effective source attribution has to satisfy some key properties: The watermarking framework should (1) achieve **accurate** source attribution, (2) be **robust** against malicious attacks on the watermarks, (3) **preserve the performance** (i.e., text generation ability) of the LLM, (4) be **scalable** to a large number of data providers, (5) ensure that the generated watermarks are **transferable** to (i.e., persist after being used as training data for) other LLMs, and (6) be **adaptable** to fit different LLMs. Sec. 2 discusses these key properties in more detail. To this end, this paper introduces a *WAtermarking for Source Attribution* (`WASA`) framework which, to our best knowledge, is the first watermarking framework capable of producing LLMs whose generated texts allow for effective source attribution. Our `WASA` framework assigns a unique watermark (i.e., imperceptible to human eyes) to every data provider, and enables an LLM (coined as `WASA`-LLM) to learn an accurate mapping from the texts of different data providers to their corresponding watermarks (Sec. 3). So, if a data provider is responsible for generating a sentence, then our `WASA`-LLM is able to include the unique watermark of this data provider in this generated sentence, which naturally supports source attribution. Our contributions are summarized below:

- We propose to use watermarking to perform source attribution (and achieve data provenance) for LLM-generated synthetic texts and identify the key properties of such watermarking frameworks.
- We introduce the `WASA` watermarking framework which satisfies these key properties and is hence capable of producing LLMs whose generated synthetic texts allow for effective source attribution.
- We perform extensive empirical evaluations (Sec. 4) to verify that our `WASA` framework satisfies these key properties and achieves effective source attribution (and data provenance).

## 2 KEY PROPERTIES OF WATERMARKING FOR SOURCE ATTRIBUTION

Here, we will discuss the key properties a watermarking framework has to satisfy to achieve effective source attribution (and hence data provenance), as well as how our `WASA` framework satisfies them.

1. **Accuracy.** Watermarks embedded into the LLM-generated texts should allow for accurate source attribution. Our `WASA` framework achieves this by enabling its trained `WASA`-LLM to learn an accurate mapping from the texts of different data providers to their corresponding watermarks. Specifically, *watermark prediction* (using texts) is made possible by training our `WASA`-LLM using watermarked texts (Sec. 3.1), and the prediction/generation spaces are separated for the texts and watermarks to both *reduce the complexity of watermark prediction* (Sec. 3.2) and *explicitly enforce watermark generation* in LLM-generated synthetic texts (Sec. 3.3). As empirically verified in Sec. 4.1, our `WASA` framework can achieve accurate source attribution.

2. **Robustness.** Watermarks should be robust against malicious attacks. Since our trained `WASA`-LLM is able to learn an accurate mapping from the texts to the watermarks as mentioned above,

This sentence is embedded with a 10-character watermark.      This sentence is not embedded with a 10-character watermark.

This sentence is embeddedU+200BU+200DU+2063U+200CU+200CU+2064U+2064U+2062U+2064U+2063 with a 10-character watermark.

Figure 2: Sentences embedded (top left) and not embedded (top right) with our imperceptible watermark visualized in the bottom sentence.

(a) it can be exploited to *regenerate* the watermarks using the (cleaned) generated texts after their watermarks are tampered with (even when the generated texts are under other additional attacks), and (b) it is still able to generate the correct watermarks even if the input texts (prompts) are perturbed. As empirically verified in Sec. 4.2, our `WASA` framework is indeed robust against these types of attacks.

3. **Scalability.** The watermarking framework should cater to a large number of data providers. Since we assign to each data provider a watermark of 10 characters with each selected among 6 Unicode characters (Sec. 3.1), we can represent over 60 million unique watermarks/data providers. Our `WASA`-LLM's ability to learn accurate texts-to-watermarks mapping ensures that its source attribution remains accurate with more data providers, as empirically verified in Sec. 4.3.

4. **Performance Preservation.** Watermarks should (a) not significantly degrade the text generation ability of the LLM (b) nor affect the readability of the LLM-generated synthetic texts too much. To preserve (a), our `WASA`-LLM only requires a *small number of watermarks* to be embedded into the training text data in order to achieve accurate source attribution, as validated in App. G.2. The watermarks are carefully designed (Sec. 3.1) to achieve (b), as shown in App. C.

5. **Transferability.** After the generated watermarked texts are used as training data for other LLMs, their generated texts should preserve the watermarks. Our `WASA` framework achieves this by ensuring that the watermarked data used to train our `WASA`-LLM has the same structure as the generated watermarked data (i.e., they both embed a 10-character watermark). This allows our generated watermarked data to be readily used as the training data for other LLMs.

6. **Adaptability.** The watermarking framework can be easily adapted to fit different LLMs. Our `WASA` framework satisfies this because it only requires mild modifications to the LLMs and can hence adopt a wide variety of LLMs using the transformer architecture. We have empirically demonstrated `WASA`'s adaptability by employing multiple LLMs in our experiments (Sec. 4.4).

We have only listed above the most essential properties of such watermarking frameworks; there may be additional considerations depending on the specific applications. In Sec. 3, we will discuss in more detail how our `WASA` framework satisfies these key properties due to our algorithmic designs.

## 3 WATERMARKING FOR SOURCE ATTRIBUTION (`WASA`) FRAMEWORK

We will first discuss in Sec. 3.1 how we design the watermarks and embed them into text data. Then, we will introduce in Sec. 3.2 how we use the watermarked texts to train our `WASA`-LLM and how its design satisfies the key properties discussed in Sec. 2. Finally, we will discuss in Sec. 3.3 how our trained `WASA`-LLM generates synthetic texts with watermarks and in Sec. 3.4 how the generated texts can be used for effective source attribution and data provenance.

### 3.1 EMBEDDING WATERMARKS INTO TEXTS FOR LLM TRAINING

To begin with, for each data provider, the LLM platform operator generates a unique watermark and then embeds the watermark into the texts from this data provider.

**Design of Watermarks.** To scale to a large number of data providers and still preserve the semantic meaning of the texts, we construct the watermarks using Unicode characters which are imperceptible to human eyes (yet can be decoded by machine learning models). Some of these invisible characters have also been adopted in other studies with language models (Boucher et al., 2022). Every watermark is made up of 10 characters, each of which is chosen among the following 6 Unicode characters: U+200B: Zero Width Space, U+200C: Zero Width NonJoiner, U+200D: Zero Width Joiner, U+2062: Invisible Times, U+2063: Invisible Separator, and U+2064: Invisible Plus. We chose these characters because they are found to be invisible on many commonly used platforms. So, these watermarks preserve the semantic meaning of the original texts to human readers (Fig. 2). Also, note that our `WASA` framework can easily adopt other choices of characters depending on the use cases. Moreover, these 10-character watermarks allow us to construct over 60 million (i.e., $6^{10}$) unique watermarks and hence achieve **scalability** to a large number of data providers. As shown in App. G.7, reducing the watermark length trades off scalability for source attribution accuracy.

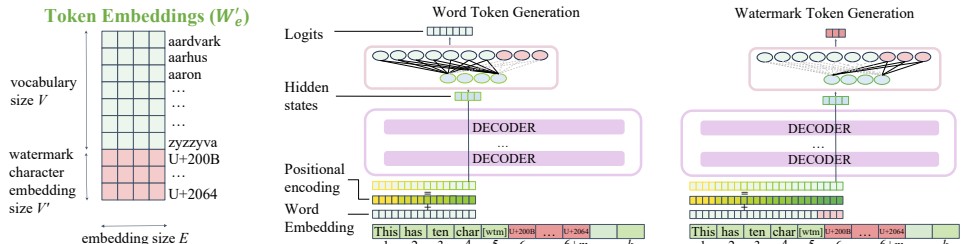

Figure 3: Separation of token embeddings and prediction spaces for texts and watermarks.

**Embedding Watermarks into Sentences.** To enable our WASA-LLM to learn the mapping from the texts of different data providers to their unique watermarks, it is important to only embed watermarks into the sentences that are *representative of the unique characteristics of the data providers*. To this end, we calculate the *term frequency-inverse document frequency* (TF-IDF) scores of all sentences from a data provider and select the sentences with the top $20\%$ of the TF-IDF scores (i.e., most representative sentences) for watermarking, which empirically yields the best trade-off of source attribution accuracy vs. text generation performance among different tested proportions, as reported in App. G.2. For every selected sentence, we embed our 10-character watermark at a random position in the sentence, which allows the LLM to learn to map texts of different lengths to the watermarks and also makes it harder for an adversary to remove/modify the watermarks. As empirically verified in App. G.5, our method of selecting sentences for watermarking based on TF-IDF indeed leads to more accurate source attribution than random selection.

## 3.2 TRAINING WASA-LLM WITH WATERMARKED TEXTS

Here, we consider the practical scenario where the LLM is already pre-trained before being used by our WASA framework, and we refer to our training of the LLM as *second-stage pre-training*. However, our framework can also be used to train an LLM from scratch without modifications.

**Preliminaries on LLMs.** Denote an unsupervised corpus by $D = [s_1, s_2, \ldots, s_n]$, in which every $s_i$ is a *block*/sequence of $k$ tokens: $s_i = [u_1, u_2, \ldots, u_k]$ where $k$ is the context window. We consider LLMs with a decoder-only structure (e.g., GPT, OPT). For a sequence $s_i$ from $D$, given a sub-sequence $s = s_i[1 : j - 1] = [u_1, \ldots, u_{j-1}]$ comprising the first $j - 1$ tokens in $s_i$, the feed-forward operations of the LLM $\mathbb{M}$ produces a predictive distribution $P(u_j)$ of $j$-th token $u_j$:

$$h_0 = s \cdot W_e + W_p, \quad h_\tau = \text{decoder}(h_{\tau-1}) \text{ for } \tau = 1, \ldots, l, \quad z = h_l[j-1] \cdot W_e^\top, \quad P(u_j) = \text{softmax}(z). \tag{1}$$

As shown in (1), given an input sequence $s$ of $j - 1$ tokens, the LLM $\mathbb{M}$ firstly uses the token embedding matrix $W_e$ to project these $j - 1$ tokens into the embedding space with an embedding/hidden dimension $E$. The dimension of $W_e$ is $V \times E$ where $V$ is the vocabulary size. Then, a positional encoding matrix $W_p$ (with dimension $(j - 1) \times E$) is added to the embedding $s \cdot W_e$ to obtain $h_0$ (1). Next, $h_0$ is sequentially passed through $l$ decoder layers to produce $h_l$ (1) such that every decoder layer consists of a multi-head self-attention layer followed by a position-wise feed-forward layer. Lastly, $\mathbb{M}$ multiplies $h_l[j - 1]$ (i.e., last hidden state of $(j - 1)$-th token) by $W_e^\top$ to produce a $V$-dimensional logit vector, after which a softmax layer is applied to produce a predictive distribution $P(u_j)$ of $j$-th token $u_j$ over the entire vocabulary of $V$ tokens (1). The training objective of an LLM is to maximize the log-likelihood $L(s_i)$ of a sequence $s_i$ of tokens, which is equivalent to minimizing the cross entropy loss $Loss(s_i)$:

$$L(s_i) = \sum_{j=2}^k \log P(u_j | u_1, \ldots, u_{j-1}), \quad Loss(s_i) = \sum_{j=2}^k \text{CE}(P(u_j | u_1, \ldots, u_{j-1}), u_j) \tag{2}$$

where $P(u_j | u_1, \ldots, u_{j-1})$ (i.e., similar to $P(u_j)$ in (1)) is the probability of $j$-th token $u_j$ conditioned on the preceding $j - 1$ tokens $[u_1, \ldots, u_{j-1}]$, and CE is the cross entropy loss.

**Forward Pass.** To ease exposition, we consider one watermark in a block, but our design also supports multiple watermarks. Our 10-character watermark forms a consecutive sub-sequence of 10 tokens. Denote a block with an embedded watermark by $s_i' = [u_1, u_2, \ldots, u_t, w_1, w_2, \ldots, w_m, u_{t+1}, \ldots, u_{k-m}]$ where $m = 10$ and the $u$'s and $w$'s are the word

and watermark tokens, respectively. There may be sequences $s_i'$'s not embedding any watermark. Hereafter, we will use $t$ to denote the token index before the first watermark token.

To begin with, we augment the original vocabulary (of size $V$) by our $V' = 6$ watermark characters (Sec. 3.1), which leads to a new vocabulary size of $V + V'$. That is, the dimension of our modified token embedding matrix $W_e'$ is $(V + V') \times E$ (Fig. 3). For a sequence $s_i'$, given a sub-sequence $s' = s_i'[1 : j - 1]$ comprising the first $j - 1$ tokens in $s_i'$ (i.e., comprising either only word tokens or both word and watermark tokens), the same feed-forward operations in (1) are applied to $s'$ (except that $W_e$ is replaced by $W_e'$) to produce $h_l$. Next, depending on whether the ground-truth $j$-th token being predicted is a word token $u$ or watermark token $w$, we adopt *two separate prediction spaces* (i.e., separate softmax layers). When predicting a *word token* $u$, the first $V$ rows of the matrix $W_e'$ are used to form the $E \times V$ matrix $(W_e'[1 : V])^\top$ as the linear layer. A softmax layer is then applied to produce the predictive distribution $P_u(u)$ of $u$ over the vocabulary of $V$ word tokens:

$$z_u = h_l[j - 1] \cdot (W_e'[1 : V])^\top, \quad P_u(u) = \text{softmax}(z_u) \, . \tag{3}$$

When predicting a *watermark token* $w$, the last $V'$ rows of $W_e'$ are used to form the $E \times V'$ matrix $(W_e'[V + 1 : V + V'])^\top$ as the linear layer. A softmax layer is then applied to produce the predictive distribution $P_w(w)$ over $V'$ watermark tokens:

$$z_w = h_l[j - 1] \cdot (W_e'[V + 1 : V + V'])^\top, \quad P_w(w) = \text{softmax}(z_w) \, . \tag{4}$$

This separation of the prediction/generation spaces of the word tokens (3) and watermark tokens (4) allows us to use *a small number of additional parameters* (i.e., $E \times V'$ instead of $E \times (V + V')$) for watermark prediction based on the hidden states of WASA-LLM. Moreover, this separation allows us to explicitly enforce the generation of watermarks (i.e., using its designated generation space) when we use the trained WASA-LLM to generate synthetic texts, as discussed in Sec. 3.3. As a result, the watermarks can be *regenerated* using cleaned texts after they are manipulated, and the correct watermarks can still be generated even if the input texts (i.e., prompts) are perturbed, hence ensuring the **robustness** of our WASA framework; more details are in Sec. 4.2.

The two separate softmax layers naturally lead to the following separate log-likelihoods:

$$L_{\text{lm}}(s_i') = \sum_{j=2}^{t} \log P_u(u_j | u_1, \ldots, u_{j-1}) + \sum_{j=t+1}^{k-m} \log P_u(u_j | u_1, \ldots, u_t, w_1, \ldots, w_m, u_{t+1}, \ldots, u_{j-1}), \tag{5}$$

$$L_{\text{wtm}}(s_i') = \sum_{j=1}^{m} \log P_w(w_j | u_1, \ldots, u_t, w_1, \ldots, w_{j-1}) \tag{6}$$

where $L_{\text{lm}}(s_i')$ (5) is the log-likelihood of word tokens, which is split into two sums before and after the watermark tokens, while $L_{\text{wtm}}(s_i')$ (6) is the log-likelihood of watermark tokens, which explicitly encourages the LLM to learn texts-to-watermarks mapping.[1] The overall log-likelihood we aim to maximize is therefore $L_{\text{WASA-LLM}}(s_i') = L_{\text{lm}}(s_i') + L_{\text{wtm}}(s_i')$. This is equivalent to minimizing the cross entropy loss $Loss_{\text{WASA-LLM}}(s_i') = Loss_{\text{lm}}(s_i') + Loss_{\text{wtm}}(s_i')$ in which

$$Loss_{\text{lm}}(s_i') = \sum_{j=2}^{t} \text{CE}(P_u(u_j), u_j) + \sum_{j=t+1}^{k-m} \text{CE}(P_u(u_j), u_j) \, , \quad Loss_{\text{wtm}}(s_i') = \sum_{j=1}^{m} \text{CE}(P_w(w_j), w_j) \tag{7}$$

represent the losses for the word and watermark tokens, respectively. For simplicity, in (7), we omit the conditioning on the preceding tokens in $P_u(u_j)$ and $P_w(w_j)$, which can be found in (5) and (6). The maximization of the log-likelihood of the watermarks conditioned on the texts (6), together with the separation of the prediction/generation spaces, enables our trained WASA-LLM to **accurately** learn the mapping from the texts to watermarks. This allows us to achieve a high **accuracy** in source attribution, which will be empirically verified in Sec. 4.1.

**Backward Pass.** Due to the separation of the softmax layers and losses for the word and watermark tokens, the backward pass for updating the parameters $W_e'$ in the last linear layer is also separated. That is, the gradients of word token loss $Loss_{\text{lm}}(s_i')$ and watermark token loss $Loss_{\text{wtm}}(s_i')$ (7) are responsible for updating $(W_e'[1 : V])^\top$ (3) and $(W_e'[V + 1 : V + V'])^\top$ (4), respectively. Specifically, the gradient update rule for $W_e'$ (with learning rate $\alpha$) can be expressed as $W_e' \leftarrow W_e' - \alpha h_l \cdot \nabla_z$ where $\nabla_z$ is a $(V + V')$-dimensional gradient vector allowing the separated gradient updates to be

---

[1]To simplify exposition, for the second sum in (5), when $j = t + 1$, the term reduces to $\log P_u(u_j | u_1, \ldots, u_t, w_1, \ldots, w_m)$. In (6), when $j = 1$, the term reduces to $\log P_w(w_j | u_1, \ldots, u_t)$.

easily achieved in a unified manner, as described below. Next, using the respective losses for word and watermark tokens (7), the gradient vectors w.r.t. $z_u$ and $z_w$ are calculated as $V$-dimensional $\nabla_{z_u} = \partial \text{CE}(P_u(u_j), u_j)/\partial z_u$ and $V'$-dimensional $\nabla_{z_w} = \partial \text{CE}(P_w(w_j), w_j)/\partial z_w$, respectively. When the loss is calculated from predicting a *word token* $u_j$ (7), let $\nabla_z = [\nabla_{z_u}, 0_{V'}]$ where $0_{V'}$ is a $V'$-dimensional all-zero vector. When the loss results from predicting a *watermark token* $w_j$ (7), let $\nabla_z = [0_V, \nabla_{z_w}]$. Note that for the parameters in the last linear layer which are responsible for predicting the *word tokens* using the hidden state (i.e., parameters $(W'_e[1:V])^\top$ in (3)), the gradient updates are *not affected by the loss for the watermark tokens*. This helps us to further limit the impact of the added watermarks on the original ability of the LLM to generate high-quality synthetic texts and hence **preserve its performance**. For the parameters in the other transformer layers (except for the frozen layers), their updates are performed using the gradients w.r.t. the losses for both the word and watermark tokens; see App. E.2 for more details.

Note that both our forward pass and backward pass only require mild modifications to an LLM. Therefore, our WASA framework can be easily adapted to fit a wide variety of LLMs, which ensures its **adaptability** property.

### 3.3 GENERATING TEXTS WITH EMBEDDED WATERMARKS USING WASA-LLM

After our WASA-LLM is trained (Sec. 3.2), it can generate synthetic texts which naturally include both the word and watermark tokens due to their *separate prediction/generation spaces*. To further improve the alignment between our training and generation stages, we introduce a *special token* $[WTM]$ which is similar to other specialized tokens and in the vocabulary of $V$ word tokens: When training our WASA-LLM using the watermarked texts, $[WTM]$ is added right before the watermark tokens during tokenization so that the presence of $[WTM]$ indicates that the subsequent $m = 10$ tokens are watermark tokens; when generating texts, if $[WTM]$ is encountered/generated, then it indicates that our WASA-LLM should switch to generating watermark tokens. Specifically, when the token $[WTM]$ is not generated, our WASA-LLM generates word tokens by applying multinomial sampling to $P_u$ (3) which is a distribution over the $V$ word tokens. When $[WTM]$ is generated, our WASA-LLM switches to generating watermark tokens by applying pure beam search to $P_w$ (4) which is a distribution over the $V'$ watermark tokens. After $m = 10$ watermark tokens have been generated, our WASA-LLM resumes the word token generation. Fig. 4 (App. C) shows the WASA-LLM-generated synthetic texts with embedded watermarks, which verifies that the watermarks are imperceptible to human eyes. However, the generated watermarks can be decoded by a *watermark decoder* algorithm introduced in Sec. 3.4 and hence used for source attribution and data provenance.

### 3.4 USING WATERMARKS FOR SOURCE ATTRIBUTION AND DATA PROVENANCE

Source attribution and data provenance can be easily performed using the synthetic texts generated by our trained WASA-LLM. When a user requests **source attribution** for some synthetic texts generated by our WASA-LLM, the LLM platform operator uses a designated *watermark decoder* algorithm to extract the generated watermark from the texts and then attribute these texts to the source (data provider) whose watermark matches the generated watermark (Fig. 1). Meanwhile, the data providers are also given both their own unique watermarks (Sec. 3.1) and the *watermark decoder* so that they can request their **data provenance**. Specifically, when a data provider wants to check its data provenance, it can firstly use its own text data (without watermark) as the input/prompt to the WASA-LLM to obtain a generated watermark, and then verify whether the generated watermark matches its own watermark (Fig. 1). Details on the matching algorithm are elaborated in App. D.

## 4 EXPERIMENTS

We perform extensive empirical evaluations to validate that our WASA framework satisfies the 6 key properties in Sec. 2. The experimental results shown below are the average taken from 5 random seeds. Following that of Clement et al. (2019), we download academic papers from ArXiv and apply post-processing to obtain a cleaned dataset (i.e., detailed in App. E.1) which we call Clean-ArXiv-Corpus (or ArXiv for short). The ArXiv dataset contains academic papers from several categories, each of which is treated as a *data provider*. We also use the BookSum dataset (Kryściński et al., 2022) consisting of various books, each of which is considered a *data provider*. We adopt 10 data providers for each dataset in our main experiments and show that our WASA can scale to a larger number (up to 100) of data providers in Sec. 4.3. We obtain WASA-LLM from our second-stage pre-training (Sec. 3.2) of the pre-trained GPT2-Large (Radford et al., 2019) or OPT-1.3B (Zhang et al.,

| | ArXiv dataset | | | | | | BookSum dataset | | | | |
| | GPT2 | | | OPT | | | GPT2 | | | OPT | |
| acc. | top-3 acc. | top-5 acc. | acc. | top-3 acc. | top-5 acc. | acc. | top-3 acc. | top-5 acc. | acc. | top-3 acc. | top-5 acc. |
|---|---|---|---|---|---|---|---|---|---|---|---|
| 74.84% | 95.76% | 98.56% | 78.36% | 99.04% | 99.36% | 77.92% | 91.80% | 96.52% | 83.20% | 93.84% | 97.80% |

Table 1: Accuracies of top-1, top-3, & top-5 source attribution (resp. denoted by 'acc.', 'top-3 acc.', & 'top-5 acc.') by WASA-LLM from 2nd-stage pre-training of different models on various datasets.

2022) model. App. E gives more details on the datasets and model training. Fig. 5 in App. E.2 shows the training convergence of our WASA-LLM in terms of the losses for word and watermark tokens.

## 4.1 ACCURACY

To facilitate easy evaluations of the source attribution **accuracy**, for each data provider, we use the sentences selected for watermarking (after removing the watermarks) as the inputs/prompts to the trained WASA-LLM, and use the watermarks embedded into the generated texts for source attribution. This simplifies the evaluations because the corresponding data provider of the input sentence is naturally the ground-truth source. Specifically, we first select 50 sentences from each data provider (i.e., 50 trials); more details are in App. F.1. Next, we use the first 200 characters of every selected sentence (without watermarks) as the input/prompt to the trained WASA-LLM which then generates synthetic texts (by continuing the sentence) with a token length of 100. The watermark embedded into the generated sentence is then used for source attribution, i.e., the source attribution is correct if this watermark matches the watermark of the data provider corresponding to this sentence (Sec. 3.4). As a result, for every data provider, the accuracy of source attribution is calculated as

$$\text{accuracy} = (\text{number of correct watermarks}) / (\text{number of trials}) . \quad (8)$$

To mitigate the impact of the length of the generated sentence on our evaluations (i.e., a watermark may not be generated if the generated sentence is too short), we use a simple technique to enforce watermark generation: If a watermark is not generated, then we force the generation of a watermark by adding the token $[WTM]$ to the end of the sentence (Sec. 3.3). This is only adopted to simplify the evaluations; as verified in App. G.3, naturally generated and forcefully generated watermarks lead to comparable source attribution accuracy. We will also empirically show in App. G.4 that this enforced watermark generation is not necessary if the generated texts are long enough.

**Top-$k$ Source Attribution.** In addition to attributing a generated sentence to a single source by using one watermark, it may be acceptable for some users to attribute a generated sentence to multiple possible sources that contain the true source. To account for these scenarios, we propose *top-$k$ source attribution* in which we modify our watermark generation (Sec. 3.3) so that when the token $[WTM]$ is encountered, we generate the top $k > 1$ watermarks with the largest beam search scores. In this case, source attribution is successful if the true watermark is contained in these $k$ watermarks, so the *top-$k$ accuracy* can be defined by replacing the number of correct watermarks in (8) with the number of generated sentences whose $k$ watermarks contain the true watermark.

**Source Attribution Accuracy.** Table 1 reports the source attribution accuracy averaged over 10 data providers which implies the accuracy of random guess is 10%. So, our WASA framework consistently achieves *accurate source attribution for both datasets and language models*; Table 6 in App. F.1 gives the source attribution accuracy for different data providers. We have also performed a fine-grained analysis of the errors incurred by our WASA framework in source attribution. Results in Table 8 (App. F.1) suggest that most source attribution errors are caused by generated texts exhibiting the characteristics of multiple data providers. This further substantiates the reliable watermark generation ability of our WASA-LLM since it almost never generates incorrect (unseen) watermarks.

**Data Provenance.** We will show here that WASA's ability to perform reliable source attribution also allows us to achieve accurate data provenance. When a data provider requests data provenance, it uses its own text data (without watermark) as the input/prompt to our trained WASA-LLM to verify whether the generated watermark matches its own (Sec. 3.4). We consider 20 categories/data providers in the ArXiv dataset, including 10 categories whose data was used for second-stage pre-training of GPT2 to obtain WASA-LLM and 10 other categories whose data was not used. We select 50 papers from each category and choose a sentence from every selected paper to use as the input/prompt to WASA-LLM for generating a watermark. The results in Table 7 (App. F.1) show that for the first 10 categories whose data was *not used* to obtain WASA-LLM, we are consistently able to recognize that their data was not misused; for the other 10 categories whose data *was used* to obtain WASA-LLM, we can also identify this with high accuracy of 74.84% and top-3 accuracy of 95.76%.

| attack strength | attacks on generated sentences with embedded watermarks | | | | | | attacks on input sentences | | | | | |
| | insertion attack | | deletion attack | | synonym substitution | | insertion attack | | deletion attack | | synonym substitution | |
| | acc. | top-3 acc. | acc. | top-3 acc. | acc. | top-3 acc. | acc. | top-3 acc. | acc. | top-3 acc. | acc. | top-3 acc. |
| 0% | 71.60% | 93.76% | 71.60% | 93.76% | 71.60% | 93.76% | 74.84% | 95.76% | 74.84% | 95.76% | 74.84% | 95.76% |
| Localized | 71.40% | 93.56% | - | - | - | - | 74.20% | 95.40% | - | - | - | - |
| 5% | 70.12% | 93.20% | 71.08% | 93.92% | 70.52% | 93.52% | 74.20% | 95.40% | 73.56% | 95.52% | 72.84% | 95.24% |
| 10% | 69.12% | 92.20% | 71.84% | 93.68% | 71.02% | 92.88% | 72.88% | 94.68% | 72.96% | 94.68% | 73.60% | 95.00% |
| 15% | 66.92% | 91.96% | 71.36% | 94.04% | 70.96% | 92.72% | 71.52% | 93.20% | 72.68% | 94.12% | 71.88% | 94.20% |
| 20% | 65.12% | 91.44% | 70.00% | 93.24% | 69.20% | 93.20% | 68.60% | 93.40% | 72.68% | 94.12% | 72.08% | 93.76% |

Table 2: Source attribution accuracy using regenerated watermarks by WASA-LLM (from 2nd-stage pre-training of GPT2 on ArXiv dataset) under various attacks on **generated sentences with embedded watermarks** (*in addition to watermark removal/modification attacks*) and on **input sentences**.

| n_categories | random guess | GPT2 | | | OPT | | |
| | | acc. | top-3 acc. | top-5 acc. | acc. | top-3 acc. | top-5 acc. |
| 10 | 10.00% | 74.84% | 95.76% | 98.56% | 78.36% | 99.04% | 99.36% |
| 25 | 4.00% | 66.48% | 90.69% | 94.05% | 69.76% | 90.48% | 95.76% |
| 50 | 2.00% | 56.44% | 80.19% | 87.54% | 61.14% | 82.63% | 89.37% |
| 100 | 1.00% | 45.06% | 68.61% | 78.76% | 48.86% | 73.34% | 81.54% |

Table 3: Source attribution accuracy for different no. of categories/data providers on ArXiv dataset.

## 4.2 ROBUSTNESS

Our WASA framework is robust against malicious attacks aiming to disrupt the source attribution via attacks on the generated watermarked sentences and on the input sentences.

**Watermark Removal/Modification Attack.** An adversary may remove/modify the watermarks in our generated sentence to sabotage the source attribution accuracy. Due to the ability of our WASA-LLM in learning an accurate texts-to-watermarks mapping, the watermark can be *regenerated* if it is manipulated. Specifically, we clean the generated sentence by removing the corrupted watermark, and use the cleaned sentence as input/prompt to WASA-LLM to regenerate the watermark (without synthetic texts) which is then used for source attribution. The regenerated watermarks by WASA-LLM (from second-stage pre-training of GPT2 on ArXiv dataset) lead to an overall accuracy (top-3 accuracy) of $71.60\%(93.76\%)$ which is comparable to the original $74.84\%(95.76\%)$ (Table 1). So, our watermark regeneration is an effective defense mechanism. Besides removing/modifying the watermark, an adversary may *additionally modify the content of the generated sentence*:

**Additional Attacks.** We also consider additional attacks on generated sentences with embedded watermarks and on input sentences, including insertion, deletion, synonym substitution, and syntactic transformation attacks. Table 2 reports the source attribution accuracy under the first 3 attacks, while App. F.2 reports that under the last attack along with all the attack descriptions. For such attacks (*in addition to watermark removal/modification attacks*) on generated sentences, watermark regeneration is used. The results show that though the attacks deteriorate attribution accuracy, high source attribution accuracy can still be preserved. This can again be explained by the reliable texts-to-watermarks mapping of our WASA-LLM, which is robust against perturbations to the input/prompt.

## 4.3 SCALABILITY

Here, we will verify WASA's ability to scale to a large number of data providers. We follow the experimental setup in Sec. 4.1 and increase the number of data providers. Results in Tables 3 and 9 (App. F.3) show that as the number of data providers increases, the source attribution accuracy inevitably decreases yet still remains relatively high compared with random guessing. With more data providers, we would recommend using $k > 1$ in top-$k$ source attribution due to a higher resulting accuracy and identifying the true source from among them.

## 4.4 OTHER KEY PROPERTIES

**Performance Preservation.** We show that our WASA framework can preserve the ability of the LLM in generating high-quality text in Table 10 (with more details in App. F.4) and ensure readability of WASA-LLM-generated synthetic text in App. C.

**Transferability.** Our generated watermarked text has *the same structure* as the watermarked text used to train our WASA-LLM: They both embed 10-character watermarks into texts with characters from the same vocabulary. So, our generated watermarked text can be readily used as training data for other LLMs that, like our WASA-LLM, can also generate synthetic text with watermarks. That is, our generated watermarked text is **transferable** to other LLMs as their training data.

**Adaptability.** Our `WASA` framework only requires mild modifications to existing LLMs (Sec. 3.2) and can hence be easily adapted to fit various LLMs. This has been verified by our results in Secs. 4.1 and 4.3 that given the same experimental setup, accurate source attributions can be achieved by `WASA`-LLM that is obtained from our second-stage pre-training of various LLMs (i.e., GPT2, OPT).

## 5  ABLATION STUDIES

We have also performed ablation studies to assess **(a)** the effectiveness of the designated embedding space for watermark tokens and separation of the prediction/generation spaces (App. G.1), **(b)** the impact of number of watermarks embedded in training data according to the percentage of top TF-IDF scores in Sec. 3.1 (App. G.2), and to reveal insights about **(c)** the enforced watermark generation in Sec. 4.1 has minimal impact on source attribution accuracy (App. G.3), **(d)** increasing the length of generated texts leads to less forcefully generated watermarks and comparable source attribution accuracy (App. G.4), **(e)** adopting TF-IDF to select sentences for embedding watermarks (Sec. 3.1) indeed leads to a better performance than random selection (App. G.5), **(f)** using more data (for every data provider) in our second-stage pre-training generally improves both the source attribution accuracy and text generation ability (App. G.6), **(g)** high source attribution accuracy can be achieved for watermarks with different lengths (App. G.7), **(h)** training for more epochs may further improve the source attribution accuracy (App. G.8), and **(i)** using fewer Unicode characters slightly decreases the top-$k$ source attribution accuracy (App. G.9). More details are deferred to App. G.

## 6  RELATED WORK

In this section, we will review the most closely related works on source attribution and data provenance to ours. App. B discusses additional related works on watermarking natural languages and text steganography, as well as recent works on watermarking language models. Some recent works attempt to obtain the data provenance of language models to verify whether a dataset has been used to train a language model. Song & Shmatikov (2019) adopt membership inference attacks, i.e., by training a shadow model to distinguish between training data vs. non-training data. However, their method is only applicable to scenarios where the model output is relatively short (e.g., a single word) and is hence not suitable for long generated texts which we consider in this work. Liu et al. (2023a) have proposed to plant backdoor triggers into the training texts for the language model so that a data contributor can verify whether its private text data has been used for training. However, this method is not robust against removals of the backdoor triggers, and the backdoor triggers may deteriorate the original text generation performance of the language model. Importantly, these works have only focused on data provenance, and *cannot be easily adapted to perform effective source attribution*. Abdelnabi & Fritz (2021) considers a different problem and proposes to embed a message into a text via adversarial training. However, their method can only embed the message *after* the text is generated by the language model and hence *cannot be used for source attribution* where the language model generates the message/watermark. Some recent works in computer vision have tackled the problem of source attribution (Marra et al., 2018; Yu et al., 2019; 2021). However, to the best of our knowledge, effective source attribution for the texts generated by language models remains an open problem which requires significantly different approaches and is the focus of our work.

## 7  CONCLUSION

This paper describes our proposed `WASA` framework which allows for effective source attribution (and hence data provenance) for LLM-generated synthetic texts (Sec. 3). Through extensive empirical evaluations (Secs. 4 and 5), we have shown that our `WASA` framework satisfies the key properties (Sec. 2) for watermarking frameworks to achieve effective source attribution and data provenance. Since our `WASA` is the first watermarking framework to achieve effective source attribution for LLM-generated texts, it faces some limitations which may call for future work. Firstly, we have only focused on scenarios where the data providers have balanced data with sufficient dissimilarities (Sec. 1). However, in some applications, the data from different data providers may be imbalanced or similar. Secondly, though we have shown that our `WASA` is robust against various adversarial attacks, it is unclear whether it is robust against more advanced/sophisticated attacks, which may be achieved through adversarial training in future work. Due to lack of space, we defer the discussion of some ethical considerations (e.g., privacy risks, data manipulation) for this work to App. A.

REPRODUCIBILITY STATEMENT

We have given the necessary details for reproducing the results in our work in this paper. Detailed descriptions of the datasets used and the experimental settings have been included in Sec. 4 and App. E, including the 5 specific random seed numbers for the experiment runs. Our code to reproduce the experiments has been included in the supplementary material.

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

## A    ETHICAL CONSIDERATIONS

Similar to other research topics on LLMs, watermarking the synthetic texts generated by LLMs for source attribution (and hence data provenance) requires a thoughtful and ethical approach due to its potential societal implications. That is, it is important to take necessary measures to avoid causing harm to certain parties. Potential risks related to our watermarking framework include the following:

- **Privacy Risks.** Watermarking can potentially reveal sensitive information about data providers, thus leading to privacy breaches or the possibility of re-identification if not handled carefully. In our `WASA` framework, only the watermark can be seen in the synthetic data, which does not directly imply personal information about the data providers. Privacy can be preserved given that the mapping from watermarks to data providers is kept confidential.
- **Chilling Effects.** Watermarking may discourage some data providers from sharing their datasets, especially if they fear potential misuse or unintended consequences of having their data linked to specific research outcomes.
- **Data Manipulation.** While watermarks are meant to be unobtrusive and our `WASA` framework has been shown to be robust against various adversarial attacks, there can be unforeseen real-world instances where malicious actors attempt to manipulate the watermark, which may lead to negative consequences such as the dissemination of altered or misleading information.

To address these potential risks, it is essential to carefully consider the ethical implications of our watermarking framework and implement measures to protect the privacy and interests of all involved parties, particularly those who are more susceptible to harm. Researchers should conduct comprehensive risk assessments and engage in transparent communication with data providers to ensure the responsible and ethical use of watermarked data. Additionally, incorporating diverse perspectives and involving vulnerable communities in the decision-making process can help identify and mitigate potential harm effectively.

## B    ADDITIONAL RELATED WORKS

In addition to the previous works discussed in Sec. 6 that are most closely related to ours, we will give a review of additional related works on watermarking natural languages and text steganography, as well as recent works on watermarking language models.

**Watermarking Natural Language/Text Steganography.**    In natural language processing, watermarking and steganography are closely related in that they both desire stealthiness and robustness. However, there are also important differences because watermarking emphasizes the importance of ownership, whereas steganography focuses on the secret communication of messages. Language watermarking is used to protect the integrity and authorship of digital texts (Kamaruddin et al., 2018; Podilchuk & Delp, 2001). Early approaches of language watermarking are mostly rule-based and make use of linguistic techniques such as synonym substitution (Topkara et al., 2006b) and sentence structure alteration (Topkara et al., 2006a) to embed watermarks while attempting to preserve the semantic meaning of the original texts. However, these approaches usually lead to deteriorated text quality and are not scalable. Some recent works have aimed to develop advanced text steganography methods using deep learning. The work of Yang et al. (2019) has utilized recurrent neural networks to automatically generate steganographic texts, and the work of Ziegler et al. (2019) has proposed to first convert the secret messages into bit strings and then map them to the cover text based on arithmetic coding with the help of GPT2 (Radford et al., 2019).

**Watermarking Language Models.**    Some recent works have proposed methods to add watermarks to language models in order to protect the IP of the models (Dai et al., 2022; Gu et al., 2023; He et al., 2022; Zhao et al., 2022). These methods allow the verification of model ownership and are hence able to protect the economic interests of model owners. Specifically, the work of He et al. (2022) has employed lexical replacement to watermark the language model output and used hypothesis testing for post-hoc model ownership verification. The work of Gu et al. (2023) has adopted backdoor attacks to embed black-box watermarks into pre-trained language models, which is achieved by using rare words as well as a combination of common words as backdoor triggers and verifying the watermarks by calculating the extraction success rate. Apart from model protection, the work of Kirchenbauer et al. (2023) has proposed to use watermarking to distinguish between human-generated and

model-generated synthetic texts, which is achieved by softly constraining the word choices when the model generates synthetic texts and using hypothesis testing to make the distinction. More recently, the work of Kuditipudi et al. (2023) has improved the method from Kirchenbauer et al. (2023) by developing a distortion-free method, which ensures that the watermarks do not change the sampling distribution of the texts. Importantly, these methods cannot be used to perform source attribution for the texts generated by language models, which we focus on in this work.

## C  GENERATED SYNTHETIC TEXTS WITH EMBEDDED WATERMARKS

We have discussed in Sec. 3.3 how our trained `WASA`-LLM can be used to generate synthetic texts with embedded watermarks. Fig. 4 below shows an example of the watermarked texts generated by our `WASA`-LLM, which shows that the generated watermarks that are embedded into the synthetic texts are indeed imperceptible to human eyes. Therefore, the readability of the synthetic texts will not be affected much.

> The complete data set of monochromatic opacities and a suite of codes to compute mean opacities and radiative accelerations (OPCD 2.1 1 ) have also been publicly released by Sea-Tek. \nTo obtain the most accurate result, we used the OPD-C++ code version 1.4.0 [20] as well as the PASCAL code version 5.2.7 [21]. We used the following method: In the first step, we calculated the maximum possible value for each pixel in the image, and then we computed the maximum possible radii per pixel from this maximum value. Then we applied the same procedure to calculate the mean radius and radiative acceleration at every pixel in the image. The results are shown in Figure 4. For example, we obtained the radii for a given position with an average of 10 pixels,

Figure 4: An example of our `WASA`-LLM-generated synthetic texts with embedded watermarks that are imperceptible to human eyes.

## D  WATERMARK MATCHING

**Exact Matching.**   In this work, we adopt exact matching to determine the correctness of the generated watermarks. That is, given a piece of synthetic text with watermarks and the corresponding ground-truth watermark, the generated watermark is correct only if they are strictly equal in string matching. In addition, in case multiple watermarks are generated in the synthetic data, all generated watermarks have to match the ground-truth watermark to affirm the correctness. The pseudocode for the matching algorithm is given below:

---
**Algorithm 1** Exact Matching

---
**Require:** Synthetic text $syn$, ground-truth watermark $wtm_g$
1: **if** $\exists\, wtm$ in $syn$ **then**
2:     $wtms \leftarrow watermark\ decoder(syn)$
3:     **if** $wtm == wtm_g$ (by string matching) **for** all $wtm$ in $wtms$ **then**
4:         return True
5:     **end if**
6: **end if**

---

**Soft Matching.**   To further improve the source attribution accuracy in some applications, we may relax the requirement of exact watermarking matching and instead attribute the generated texts to the data provider whose watermark has the smallest Levenshtein distance to the generated watermark. However, in all our experiments, our `WASA` is able to achieve accurate source attribution without soft matching.

## E  DETAILED EXPERIMENTAL SETUP

### E.1  DATASETS

To simulate different data providers with unique characteristics, we create the Clean-ArXiv-Corpus (or ArXiv for short) dataset which consists of academic papers from ArXiv. The dataset contains

academic papers from various sub-disciplines, including computer science, physics, mathematics, public health, and other related fields. We make use of the provided metadata from the work of Clement et al. (2019) to download the corresponding PDF files and retrieve the categorization information associated with each article. Subsequently, we employ GROBID (Lopez, 2008–2023) to parse and extract the main body of the papers, excluding the abstract and reference sections. Our Clean-ArXiv-Corpus dataset covers a comprehensive collection of 100 distinct categories, each comprising a number of papers ranging from 2827 to 2984. We treat *every category as a data provider*, so one data provider/category is the source of each piece of text. Our main experiments in Sec. 4 are conducted using 10 categories (i.e., data providers) and we use 33% of papers from each category due to computational constraints. However, in our ablation study (App. G.6), we have also tested utilizing more data from every data provider (including 100% of the data), which has led to further improved performances and consistent conclusions. For each of the 10 categories, we further randomly split its data into training and evaluation datasets with a ratio of 9 : 1 according to the seed number. In our ablation study, we will use more categories and also use all papers in each category. More detailed information about the full Clean-ArXiv-Corpus dataset, including all 100 categories and all papers in each category, is shown in Table 4; Table 4 shows an instance of the random split into training and evaluation datasets based on seed number 2023.

In addition to the Clean-ArXiv-Corpus dataset, we also adopt the BookSum dataset (Kryściński et al., 2022). This dataset contains documents from the literature domain including novels, plays, and stories. The BookSum dataset contains 181 books and we treat *every book as a data provider*. For every data provider (i.e., book), we adopt all the text data from the book in all our experiments. More information on the BookSum dataset is shown in Table 5; Table 5 shows an instance of the random split into training and evaluation datasets based on seed number 2023.

|  | Training | Evaluation |
| --- | --- | --- |
| Papers | 264K | 29K |
| Unique tokens | 17.1M | 3M |
| Unique tokens per Category | 407K | 87K |
| Total tokens | 1.8B | 203M |
| Total tokens per Category | 18.2M | 2M |

Table 4: Information on the Clean-ArXiv-Corpus (or ArXiv for short) dataset.

|  | Training | Evaluation |
| --- | --- | --- |
| Books | 161 | 20 |
| Unique tokens | 413K | 106K |
| Unique tokens per Book | 91K | 20K |
| Total tokens | 33M | 4.6M |
| Total tokens per Book | 3.3M | 467K |

Table 5: Information on the BookSum dataset.

## E.2 EXPERIMENTAL SETTING

In our experiments, we build our WASA-LLM based on the open-source pre-trained GPT2-Large model (Radford et al., 2019) and OPT-1.3B model (Zhang et al., 2022). Based on the pre-trained weights, we perform our second-stage pre-training (Sec. 3.2) of the pre-trained GPT2-Large model or the OPT-1.3B model on the watermarked (Sec. 3.1) text data for one epoch to obtain WASA-LLM. We find that training for one epoch already allows our WASA framework to achieve compelling performances, as shown in our experiments in Sec. 4. We have also tested more training epochs in App. G.8 and the results suggest that our performances can potentially be further improved with more training epochs. We plot the convergence of the training of our WASA-LLM in terms of the losses for the word and watermark tokens in Fig. 5, which shows that our second-stage pre-training effectively reduces both losses. Importantly, the watermark token loss rapidly declines after a small number of steps, which suggests that our WASA-LLM can quickly learn an accurate texts-to-watermarks mapping.

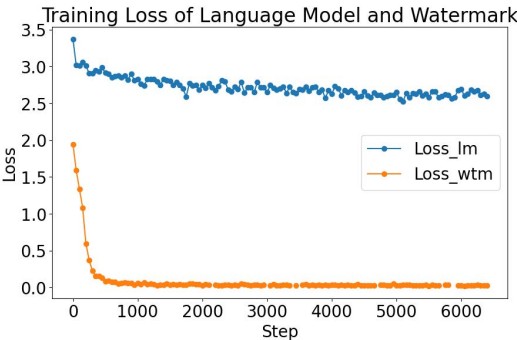

Figure 5: Training losses for word tokens (Loss_lm) and watermark tokens (Loss_wtm) when obtaining `WASA`-LLM from second-stage pre-training of the GPT2 model on ArXiv dataset.

Here, we give more details on the hyperparameters we adopted. We fix 5 seed numbers at 2021, 2022, 2023, 2024, and 2025 for obtaining reproducible results, and the results shown in this work are the average taken across that from the 5 seeds. We adopt the Adam optimizer with a learning rate of $5 \times 10^{-5}$ and no weight decay. We make use of the fp16 technique and a gradient accumulation of 8 to speed up training. We also adopt a gradient checkpoint to reduce memory usage so that batch size can be slightly increased. We use a block size of 512 and a batch size of 3 for most of the experiments and a batch size of 16 in the experiments to evaluate scalability. To further preserve the ability of the original pre-trained LLM models, during the second-stage pre-training, we freeze the first 12 layers of GPT2-Large (among a total of 36 layers) and freeze the first 8 layers of OPT-1.3B (among a total of 24 layers). When generating the synthetic texts (see Sec. 3.3), we use the multinomial sampling of top-60 with temperature = 0.7. We also make use of a 1.2 repetition penalty and a 2.0 length penalty to generate better synthetic data. The generation of watermarks for our `WASA`-LLM adopts a pure beam search, as discussed in Sec. 3.3, with a beam size of 5. For the baseline models (GPT2-Large and OPT-1.3B), watermark generation is performed in the same way as text generation, so we use the same hyperparameters as that specified in the baseline models. All second-stage pre-training is performed using NVIDIA RTX A5000 and A100. In our implementation, we adopt the GROBID library to process the PDF files. For model training, we adopt the Hugging Face Trainer pipeline which embeds necessary tricks to speed up the training process. The open-source GPT2-Large and OPT-1.3B are also adopted from Hugging Face.[2]

## F  MORE EXPERIMENTAL RESULTS

### F.1  ACCURACY

**More Details on Experimental Setup.** In our experiments on the source attribution accuracy, for the ArXiv dataset, we select 50 papers from each of the 10 categories (App. E.1) and for every selected paper, we choose the first sentence that has been selected for watermarking (to obtain our `WASA`-LLM from second-stage pre-training of various pre-trained LLMs, see Sec. 3.1 for more details on how we select the sentences for watermarking) as well as contains at least 200 characters. Similarly, for every book (i.e., data provider) in the BookSum dataset, we select the first 50 sentences that have been selected for watermarking as well as have at least 200 characters. As a result, for both datasets, we have selected 50 sentences to be used as the inputs/prompts to our `WASA`-LLM, which corresponds to 50 trials of source attribution for each of the 10 data providers.

**Source Attribution Accuracy for Each Data Provider.** Table 6 shows the detailed results on source attribution accuracy for the 10 different data providers, in addition to Table 1 in Sec. 4.1. The results show that the accuracy remains balanced across the data providers.

**Data Provenance.** Table 7 displays the detailed results for the experiment on data provenance presented in Sec. 4.1 of the main paper. The insights/conclusions drawn from these results have been

---

[2]`https://huggingface.co/GPT2-Large` and `https://huggingface.co/facebook/OPT-1.3B`.

| ArXiv dataset | | | | | BookSum dataset | | | | |
|---|---|---|---|---|---|---|---|---|---|
| Data Provider | GPT2 | | OPT | | Data Provider | GPT2 | | OPT | |
| | acc. | top-3 acc. | acc. | top-3 acc. | | acc. | top-3 acc. | acc. | top-3 acc. |
| hep-th | 65.60% | 94.40% | 67.60% | 99.20% | Adam Bede | 82.40% | 95.60% | 85.20% | 96.00% |
| hep-ph | 85.20% | 96.80% | 87.60% | 98.80% | David Copperfield | 80.00% | 88.40% | 77.20% | 91.60% |
| quant-ph | 74.80% | 91.60% | 76.80% | 98.00% | Dracula | 66.80% | 86.00% | 71.60% | 91.60% |
| astro-ph | 86.40% | 94.40% | 86.00% | 98.40% | Hamlet | 91.20% | 96.80% | 97.60% | 99.20% |
| cs.CV | 82.00% | 95.20% | 85.20% | 99.20% | Henry IV Part 1 | 90.40% | 98.40% | 97.20% | 99.60% |
| cs.LG | 77.60% | 98.80% | 83.20% | 99.60% | Ivanhoe | 83.60% | 94.40% | 89.20% | 93.60% |
| cond-mat.mes-hall | 64.80% | 98.40% | 74.00% | 99.20% | Jane Eyre | 74.00% | 90.00% | 80.00% | 96.40% |
| gr-qc | 76.40% | 96.40% | 82.00% | 99.20% | Little Women | 85.60% | 94.00% | 94.00% | 98.00% |
| cond-mat.mtrl-sci | 64.80% | 95.20% | 71.60% | 99.20% | Middlemarch | 72.80% | 94.40% | 76.00% | 93.20% |
| cond-mat.str-el | 70.80% | 96.40% | 69.60% | 99.60% | The Pickwick Papers | 52.40% | 80.00% | 64.00% | 79.20% |
| Overall | 74.84% | 95.76% | 78.36% | 99.04% | Overall | 77.92% | 91.80% | 83.20% | 93.84% |

Table 6: Source attribution accuracy achieved by our `WASA`-LLM (i.e., obtained from second-stage pre-training of different models on various datasets) for different data providers.

discussed in Sec. 4.1: Due to its ability to perform accurate source attribution, our `WASA` framework can also achieve reliable data provenance.

| category | n_watermark | data provenance (n_match) |
|---|---|---|
| cond-mat.soft | 50 | ✗ (0) |
| q-bio.PE | 50 | ✗ (0) |
| cs.SY | 50 | ✗ (0) |
| eess.IV | 50 | ✗ (0) |
| hep-ex | 50 | ✗ (0) |
| math.LO | 50 | ✗ (0) |
| math.NA | 50 | ✗ (0) |
| math.ST | 50 | ✗ (0) |
| nlin.SI | 50 | ✗ (0) |
| physics.class-ph | 50 | ✗ (0) |
| hep-th | 50 | ✓ (32.8) |
| hep-ph | 50 | ✓ (42.6) |
| quant-ph | 50 | ✓ (37.4) |
| astro-ph | 50 | ✓ (43.2) |
| cs.CV | 50 | ✓ (41.0) |
| cs.LG | 50 | ✓ (38.8) |
| cond-mat.mes-hall | 50 | ✓ (32.4) |
| gr-qc | 50 | ✓ (38.2) |
| cond-mat.mtrl-sci | 50 | ✓ (32.4) |
| cond-mat.str-el | 50 | ✓ (35.4) |

Table 7: Reliable data provenance can be achieved due to the ability of `WASA`-LLM to perform accurate source attribution. `WASA`-LLM is obtained from second-stage pre-training of the GPT2 model on the ArXiv dataset. Note that the numbers shown here are the average taken across 5 runs with different random seeds.

**Fine-grained Analysis of Source Attribution Errors.** To gain more insights into the behavior of our `WASA` framework, we perform a more fine-grained analysis of the errors incurred by our `WASA` framework in source attribution on the ArXiv dataset (i.e., corresponding to the results in Table 1). In Table 8, for every category (i.e., data provider), we separate the source attribution errors into two types of errors: (a) *misclassification* in which the generated watermark matches the watermark of another incorrect category, and (b) *incorrect watermark* in which the generated watermark does not match the watermark of any category. The results in Table 8 show that the vast majority of our errors result from misclassification and our `WASA`-LLM rarely generates incorrect watermarks not belonging to any category. This implies that our source attribution errors are mostly caused by the generated synthetic texts (conditioned on a sentence from a data provider/category) exhibiting the characteristics of multiple data providers. These results further substantiate the strong and reliable watermark generation ability of our `WASA`-LLM because it almost never generates incorrect (unseen) watermarks.

| category | n_watermark | n_match | misclassification | incorrect watermark |
|---|---|---|---|---|
| hep-th | 50 | 32.8 | 17.2 | 0 |
| hep-ph | 50 | 42.6 | 7.4 | 0 |
| quant-ph | 50 | 37.4 | 12.6 | 0 |
| astro-ph | 50 | 43.2 | 6.8 | 0 |
| cs.CV | 50 | 41.0 | 9.0 | 0 |
| cs.LG | 50 | 38.8 | 11.2 | 0 |
| cond-mat.mes-hall | 50 | 32.4 | 17.6 | 0 |
| gr-qc | 50 | 38.2 | 11.8 | 0 |
| cond-mat.mtrl-sci | 50 | 32.4 | 17.6 | 0 |
| cond-mat.str-el | 50 | 35.4 | 14.6 | 0 |
| Total | 500 | 374.2 | 125.8 | 0 |

Table 8: Error analysis of watermarks incurred by our `WASA`-LLM that is obtained from second-stage pre-training of the GPT2 model on the ArXiv dataset. Note that the numbers shown here are the average taken across 5 runs with different random seeds.

## F.2    ROBUSTNESS

### F.2.1    ADDITIONAL ATTACKS ON GENERATED SENTENCES WITH EMBEDDED WATERMARKS

As discussed in Sec. 4.2, an adversary may *additionally modify the content of the generated sentence* while removing/modifying the generated watermarks. Here, we will consider insertion, deletion, synonym substitution, and syntactic transformation attacks. In **insertion attacks** on a generated watermarked sentence, either one word is randomly inserted into the sentence (i.e., *localized insertion attacks*), or various words are randomly interspersed throughout the sentence (i.e., *dispersed insertion attacks*) (Kamaruddin et al., 2018). For dispersed insertion attacks, we vary the attack strengths by changing the number of inserted words from $5\%$ to $20\%$ of the total number of words in the sentence. In **deletion attacks**, some words in the text are randomly deleted. In **synonym substitution attacks** (Kamaruddin et al., 2018), an adversary substitutes some words in the generated sentence with their synonyms while preserving the semantic meaning of the sentence. We again test different attack strengths by varying the percentage of randomly deleted and substituted words. In addition, we also performed the **syntactic transformation attack** on the generated sentences whereby an adversary transforms the sentences (without altering their semantic meanings) via techniques such as modifying the prepositions, tenses, and other syntax components. Here, we adopt a strong variant of such attacks, which paraphrases the input sentence using the PEGASUS model fine-tuned for paraphrasing (Zhang et al., 2020). The accuracy (top-3 accuracy) after the syntactic transformation attacks is $66.28\%$ ($89.56\%$). The **robustness** of our `WASA` framework can be validated by the marginal performance degradation in Table 2.

### F.2.2    ATTACKS ON INPUT SENTENCES (PROMPTS)

An adversary may also manipulate the input sentence (prompt) to our trained `WASA`-LLM to disrupt watermark generation and hence source attribution. The **insertion, deletion, and syntactic transformation attacks** are the same as those described in App. F.2.1, except that these attacks are performed on the input sentences here. Similar to App. F.2.1, we vary the attack strengths for these three types of attacks. The results in Table 2 show that these attacks also only lead to marginal degradation in the source attribution accuracy. Moreover, under the strong syntactic transformation attacks, the source attribution remains accurate (with an accuracy of $63.00\%$ and a top-3 accuracy of $89.00\%$), which provides further evidence for the robustness of our `WASA` framework against attacks on the input sentences. Its robustness against these attacks can again be explained by its reliable texts-to-watermarks mapping, which allows our `WASA`-LLM to consistently generate the correct watermarks even if the prompt is perturbed.

## F.3    SCALABILITY

In Sec. 4.3, we have verified `WASA`'s scalability to a large number of data providers using the ArXiv dataset. Here, we will also show in Table 9 the source attribution accuracy for a larger number of books (i.e., data providers) using the BookSum dataset. Note that `WASA`'s scalability using the

BookSum dataset is worse than that using the ArXiv dataset because each data provider in the former offers much less data.

| n_books | random guess | GPT2 | | | OPT | | |
|---|---|---|---|---|---|---|---|
| | | acc. | top-3 acc. | top-5 acc. | acc. | top-3 acc. | top-5 acc. |
| 10 | 10.00% | 77.92% | 91.80% | 96.52% | 83.20% | 93.84% | 97.80% |
| 25 | 4.00% | 52.69% | 68.80% | 75.33% | 64.04% | 76.85% | 83.71% |
| 50 | 2.00% | 45.18% | 62.23% | 67.63% | 54.17% | 70.01% | 76.79% |
| 100 | 1.00% | 18.50% | 40.15% | 44.52% | 24.01% | 55.70% | 63.31% |

Table 9: Source attribution accuracy for different numbers of books (i.e., data providers) in the BookSum dataset.

### F.4 Performance Preservation

Here, we will show that our WASA-LLM preserves the text generation ability of the original LLM by comparing it with the original GPT2-Large model which we denote as *originalGPT*. We apply our second-stage pre-training to originalGPT using the same (but un-watermarked) data from the ArXiv dataset as that used for second-stage pre-training of the GPT2-Large model to obtain our WASA-LLM. We assess the text generation performance using several commonly used evaluation metrics (with a separate evaluation dataset, as explained in App. E.1): perplexity (i.e., lower is better), and distinct-1 and distinct-2 scores (i.e., higher is better) (Li et al., 2016). The results in Table 10 show that the text generation ability of our WASA-LLM is comparable to that of originalGPT, which indicates that our WASA framework preserves the ability of the LLM to generate high-quality texts (Sec. 2).

| models | perplexity | distinct-1 | distinct-2 |
|---|---|---|---|
| originalGPT | 12.2818 | 0.8141 | 0.9796 |
| WASA-LLM | 12.6570 | 0.8194 | 0.9800 |

Table 10: Comparison of the text generation performances achieved by our WASA-LLM (obtained from second-stage pre-training of the GPT2-Large model) vs. the baseline model on the ArXiv dataset.

## G    Detailed Results from Ablation Studies

Here, we will present detailed results from our ablation studies. In all our ablation studies, we use second-stage pre-training of the GPT2-Large model on the ArXiv dataset to obtain WASA-LLM. Note that we fix 5 seed numbers at 2021, 2022, 2023, 2024, and 2025 for obtaining reproducible results, and the results shown are the average taken across that from the 5 seeds.

### G.1    Effectiveness of our WASA-LLM Training

We have mainly implemented two important algorithmic designs to help our WASA-LLM learn an accurate texts-to-watermarks mapping (Sec. 3.2): (1) using a designated embedding space for watermark tokens and (2) separating the prediction/generation spaces for the word and watermark tokens. Here, we compare our WASA-LLM with two baselines: *tokenizerGPT* implementing only the first design of a designated embedding space for watermark tokens, and *originalGPT* (original GPT2-Large) implementing neither design. We apply our second-stage pre-training to both baselines using the same (watermarked) data from the ArXiv dataset which was used for second-stage pre-training of the GPT2-Large model to obtain our WASA-LLM, and evaluate the source attribution accuracy following that of Sec. 4.1. The results in Table 11 show that the first design alone does not improve the source attribution accuracy whereas the combination of both designs brings about a significant improvement. This is because merely creating the embedding space for watermark tokens does not help in learning the mapping from the texts to watermarks, and it is of particular importance to

combine both designs for our WASA-LLM to perform well. Moreover, our WASA-LLM achieves a significantly better source attribution accuracy at the expense of incurring more computational time. Note that *originalGPT* takes longer training time than *tokenizerGPT* because there is no designated embedding space for watermark tokens in *originalGPT*, hence resulting in more training instances used.

| model | n_watermark | acc. | n_training_samples | training time |
|---|---|---|---|---|
| RandomGuess | - | 10.00% | - | - |
| originalGPT | 412 | 45.69% | 163507 | 6h30m3s |
| tokenizerGPT | 439 | 44.01% | 140599 | 5h3m6s |
| WASA-LLM | 448 | 74.84% | 159387 | 8h09m24s |

Table 11: Comparison of source attribution accuracy achieved by WASA-LLM (obtained from second-stage pre-training of the GPT2 model) vs. the baseline models on the ArXiv dataset where 'acc.' denotes the source attribution accuracy. RandomGuess incurs an accuracy of $10\%$ since there are 10 categories.

## G.2 IMPACT OF NUMBER OF WATERMARKS IN TRAINING DATA

Here, we will evaluate the impact of the number of watermarks in the training data on the source attribution accuracy achieved by WASA-LLM. Following that of Sec. 3.1, we vary the percentage of sentences selected for watermarking (i.e., top $X\%$ of the TF-IDF scores) and evaluate its impact on our WASA-LLM obtained from second-stage pre-training of the GPT2 model on the ArXiv dataset. Fig. 6 (left) shows that as the number of watermarks increases, the source attribution accuracy firstly increases and then declines. This is because an overly small number of watermarks results in insufficient data for learning an accurate texts-to-watermarks mapping; meanwhile, if watermarks are added to an excessively large number of sentences, then some of the watermarked sentences *may not be representative of the texts from their data providers*, which also increases the difficulty of learning the mapping from the texts of the data providers to their unique watermarks (see Sec. 3.1). In addition, Fig. 6 (right) shows that increasing the number of added watermarks in general leads to worse text generation performances (i.e., larger perplexity) of the WASA-LLM. The detailed results are provided in Table 12. Moreover, Fig. 7 shows a clearer visualization of the results in smaller percentages.

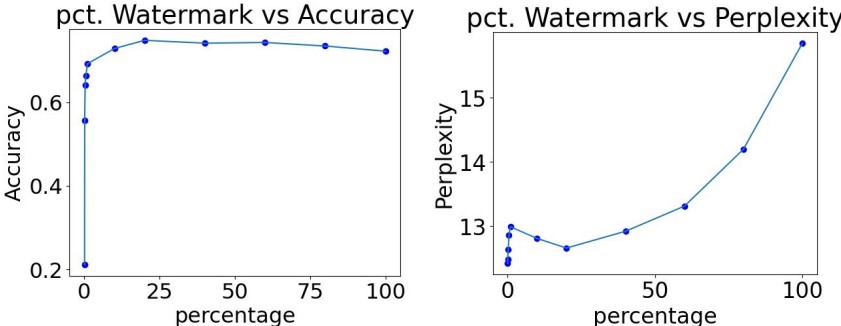

Figure 6: Source attribution accuracy and perplexity achieved by WASA-LLM (i.e., obtained from second-stage pre-training of the GPT2 model on the ArXiv dataset) vs. percentage of watermarked sentences in the training data.

## G.3 IMPACT OF ENFORCED WATERMARK GENERATION

As discussed in Sec. 4.1, to evaluate the source attribution accuracy in our experiments, we have adopted a simple technique to enforce watermark generation in order to simplify the evaluations. That is, if a watermark is not generated after the generation of the sentence is completed, we add the token $[WTM]$ to the end of the sentence to enforce the watermark generation. Here, we will evaluate the impact of this enforced watermark generation. The results in Table 13 show that the

| pct. sentences | pct. blocks | acc. | top-3 acc. | perplexity |
|---|---|---|---|---|
| 20% | 88.25% | 74.84% | 95.76% | 12.6570 |
| 40% | 96.88% | 74.16% | 95.45% | 12.9180 |
| 60% | 98.86% | 74.32% | 95.04% | 13.3096 |
| 80% | 99.38% | 73.48% | 95.40% | 14.1952 |
| 100% | 100.00% | 72.24% | 95.00% | 15.8465 |

Table 12: Comparison of source attribution accuracy achieved by WASA-LLM (i.e., obtained from second-stage pre-training of the GPT2 model on the ArXiv dataset) for different percentages of watermarked sentences in the training data. The percentage of blocks that are watermarked is given as well.

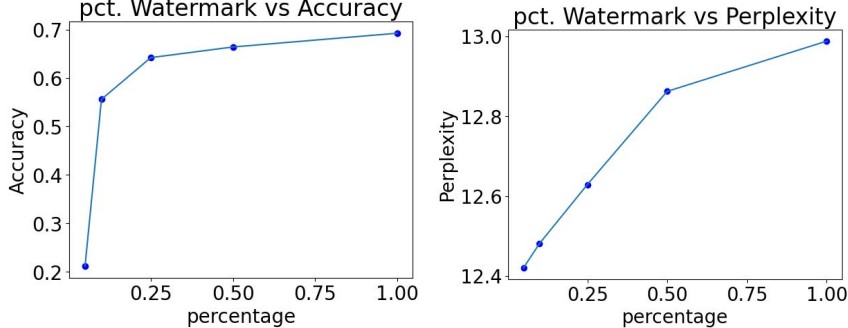

Figure 7: Source attribution accuracy and perplexity achieved by WASA-LLM (i.e., obtained from second-stage pre-training of the GPT2 model on the ArXiv dataset) vs. percentage of watermarked sentences in the training data on a smaller scale of $0.05\% - 1\%$ for a clearer visualization.

forcefully generated watermarks and naturally generated watermarks have comparable source attribution accuracy. This shows that the technique of enforced watermark generation we have adopted has minimal impact on the evaluations of the source attribution accuracy (Sec. 4.1).

| category | n_watermark_nf | n_match_nf | acc._nf | n_watermark_f | n_match_f | acc._f |
|---|---|---|---|---|---|---|
| hep-th | 45.8 | 30.4 | 66.38% | 4.2 | 2.4 | 57.14% |
| hep-ph | 44.2 | 37.8 | 85.52% | 5.8 | 4.8 | 82.76% |
| quant-ph | 46.0 | 35.4 | 77.00% | 4 | 2 | 50.00% |
| astro-ph | 44.2 | 38.6 | 87.33% | 5.8 | 4.6 | 79.31% |
| cs.CV | 44.2 | 36.4 | 82.35% | 5.8 | 4.6 | 79.31% |
| cs.LG | 44.4 | 35.0 | 78.83% | 5.6 | 3.8 | 67.86% |
| cond-mat.mes-hall | 44.8 | 28.8 | 64.29% | 5.2 | 3.6 | 69.23% |
| gr-qc | 43.2 | 33.8 | 78.24% | 6.8 | 4.4 | 64.71% |
| cond-mat.mtrl-sci | 46.6 | 30.6 | 65.67% | 3.4 | 1.8 | 52.94% |
| cond-mat.str-el | 44.6 | 31.6 | 70.85% | 5.4 | 3.8 | 70.37% |
| Total | 448 | 338.4 | 75.54% | 52 | 35.8 | 68.85% |

Table 13: Source attribution accuracy achieved by WASA-LLM (i.e., obtained from second-stage pre-training of the GPT2 model on the ArXiv dataset) for naturally generated watermarks (denoted by 'watermark_nf') vs. forcefully generated watermarks (denoted by 'watermark_f').

## G.4 IMPACT OF LENGTHS OF CONDITIONED SENTENCE AND SYNTHETIC SENTENCE

Recall that in our main experiments, we have used a sentence with 200 characters as the input/prompt (i.e., the conditioned sentence) to our WASA-LLM, and let the WASA-LLM generate synthetic texts with 100 tokens (Sec. 4.1). In this section, we vary the character lengths of both the conditioned sentence and the generated synthetic texts, and evaluate their impact on the source attribution accuracy achieved by WASA-LLM (i.e., obtained from second-stage pre-training of the GPT2 model on the ArXiv dataset). The results in Table 14 show that longer conditioned sentences (i.e., in-

puts/prompts) lead to better performances. Moreover, when the length of the conditioned sentences is fixed (at 200), increasing the length of the generated synthetic texts consistently reduces the number of forcefully generated watermarks (App. G.3) while preserving the source attribution accuracy achieved by WASA-LLM.

| len. cond. sentence | tokens syn. sentence | acc. | top-3 acc. | pct. watermark_f |
|---|---|---|---|---|
| 100 | 100 | 63.92% | 89.96% | 15.2% |
| 100 | 200 | 64.36% | 89.48% | 5.2% |
| 200 | 100 | 74.84% | 95.76% | 8.6% |
| 200 | 200 | 75.20% | 95.64% | 4.2% |
| 200 | 300 | 74.24% | 95.40% | 2.2% |
| 200 | 400 | 74.60% | 95.24% | 1.0% |

Table 14: Impact of the lengths of the conditioned sentences (inputs/prompts) and the generated synthetic sentences on the source attribution accuracy achieved by WASA-LLM (obtained from second-stage pre-training of the GPT2 model on the ArXiv dataset) where 'len. cond. sentence' stands for the character length of the conditioned sentences, 'tokens syn. sentence' refers to the number of tokens in the generated synthetic sentences, and 'pct. watermark_f' denotes the percentage of forcefully generated watermarks.

## G.5 STRATEGY FOR SELECTING SENTENCES TO WATERMARK

As we have discussed in Sec. 3.1, for every data provider, we embed watermarks into the sentences with top TF-IDF scores and then use these watermarked sentences for the second-stage pre-training (Sec. 3.2) of the GPT2 model to obtain our WASA-LLM. This is because the sentences with high TF-IDF scores are more representative of the text data from a data provider, which makes it easier to learn the mapping from the texts of different data providers to their corresponding unique watermarks. Here, we will evaluate whether this strategy is effective by comparing it with the natural baseline of randomly selecting sentences to embed watermarks. The results in Table 15 show that when selecting 20% of the sentences for watermarking, the strategy of random embedding significantly decreases the source attribution accuracy, which validates the effectiveness of our strategy of selecting sentences with high TF-IDF scores to watermark.

| embedding strategy | acc. | top-3 acc. |
|---|---|---|
| TF-IDF (ours) | 74.84% | 95.76% |
| Randomly Embed | 71.40% | 94.48% |

Table 15: Source attribution accuracy achieved by WASA-LLM (obtained from second-stage pre-training of the GPT2 model on the ArXiv dataset) using different strategies to select the sentences for watermarking.

| dataset size | acc. | top-3 acc. | perplexity |
|---|---|---|---|
| 10%: 100MB | 68.80% | 94.10% | 14.6135 |
| 33%: 300MB | 74.84% | 95.76% | 12.6570 |
| 66%: 600MB | 76.28% | 95.88% | 11.6749 |
| 100%: 1GB | 78.48% | 95.80% | 11.3171 |

Table 16: Comparison of source attribution accuracy and perplexity achieved by WASA-LLM (obtained from second-stage pre-training of the GPT2 model on the ArXiv dataset) across different dataset sizes.

## G.6 IMPACT OF AMOUNT OF DATA FOR SECOND-STAGE PRE-TRAINING TO OBTAIN WASA-LLM

Here, we will evaluate the impact of using varying amounts of data from the ArXiv dataset for our second-stage pre-training (Sec. 3.2) of the GPT2 model to obtain WASA-LLM. As discussed in App. E.1, in our main experiments for the ArXiv dataset, we have used 33% of text data from every category (i.e., data provider) to reduce computations. Here, we will vary this percentage to evaluate its impact on both the source attribution accuracy and the text generation performance achieved by our WASA-LLM. The results in Table 16 demonstrate that as more data is used, both the source attribution accuracy and the text generation ability (i.e., perplexity) achieved by our WASA-LLM are generally improved.

## G.7 IMPACT OF LENGTH OF WATERMARK

In our main experiments, we have adopted a watermark design that consists of 10 characters/tokens (Sec. 3.1). However, our WASA framework allows for the use of watermarks with different lengths. Here, we will test the impact of the length of the watermarks on the source attribution accuracy achieved by WASA-LLM (obtained from second-stage pre-training of the GPT2 model on the ArXiv dataset). The results in Table 17 show that for watermarks with 5, 10, and 15 characters, their source attribution accuracies are comparable while the 5-character watermark achieves slightly better performances. This is likely because when the watermark is shorter, the resulting watermark prediction problem becomes relatively easier (i.e., the number of parameters in the last linear layer is smaller), which may lead to better watermark prediction and generation. However, note that a long watermark is favored when there is a need to scale to a large number of data providers. Therefore, our WASA framework offers the flexibility to choose watermarks with different lengths, and the preferred watermark length can be application-dependent.

| len. watermarks | acc. | top-3 acc. |
|---|---|---|
| 5 characters | 76.12% | 95.48% |
| 10 characters | 74.84% | 95.76% |
| 15 characters | 74.12% | 95.28% |

Table 17: Source attribution accuracy achieved by WASA-LLM (obtained from second-stage pre-training of the GPT2 model on the ArXiv dataset) using watermarks with different lengths.

| n_epochs | acc. | top-3 acc. |
|---|---|---|
| 1 | 74.84% | 95.76% |
| 2 | 76.96% | 96.00% |
| 3 | 75.88% | 95.88% |

Table 18: Source attribution accuracy achieved by WASA-LLM (obtained from second-stage pre-training of the GPT2 model on the ArXiv dataset) after training with more epochs.

## G.8 IMPACT OF NUMBER OF TRAINING EPOCHS

As we have discussed in App. E.2, we have trained our WASA-LLM for one epoch during the second-stage pre-training (Sec. 3.2). Here, we will evaluate the performance of WASA-LLM after training with more epochs. The results in Table 18 show that training with multiple epochs in general further improves the performance. This demonstrates the potential of our WASA framework to achieve even better source attribution accuracy (than those presented in our current experiments) with more computations.

## G.9 IMPACT OF NUMBER OF WATERMARK CHARACTERS

In our main experiments, we have used 6 invisible Unicode characters to form each character in the 10-character watermark. Our WASA framework also allows for the use of watermarks such that each character in the watermark can be chosen among a different number of available characters. Table 19 shows the source attribution accuracy achieved by WASA-LLM (obtained from second-stage pre-training of the GPT2 model on the ArXiv dataset) when each character in the watermark can be chosen among only 2 available characters: U+200B: Zero Width Space and U+200C: Zero Width NonJoiner. The results are comparable while the one with 2 available characters shows slightly worse top-3 accuracy. This is likely because when fewer available characters are used, the watermarks for different categories are more similar to each other, which may make top-3 classification more difficult.

| n_available_characters | acc. | top-3 acc. |
|---|---|---|
| 2 | 75.48% | 89.92% |
| 6 | 74.84% | 95.76% |

Table 19: Impact of the number of available characters (used to make up each character in the 10-character watermark) on the source attribution accuracy achieved by WASA-LLM (obtained from second-stage pre-training of the GPT2 model on the ArXiv dataset).

## G.10 Effectiveness of Evaluation

In our experiment design, we assign the ground truth source of each generated text to be identical to that of the prompt sentence. Here, we would like to verify that our method of using the source of the prompt sentence as the ground truth source for the generated sentence is indeed a reliable approach, in addition to its benefit of simplifying the experimental evaluation.

A natural and reliable method to find the ground truth source of a generated text is to consult the opinion of human experts. Therefore, we would like to show that our method to determine the ground truth source is an accurate approximation to human evaluations. To avoid the substantial costs and resources associated with human evaluators, we have employed GPT-4, noted for its human-level performance across various benchmarks (OpenAI, 2023), as a surrogate 'human-like labeler'. Then, we examine whether the ground truth source determined by our method (i.e., using the source of the prompt sentence) aligns well with those determined by GPT-4. Specifically, we use GPT-4 to categorize generated texts into one of the ten ArXiv categories (i.e., data providers) using a carefully constructed prompt, as shown in Table 20. After evaluating 500 generated texts, we have found that 89.6% of GPT-4's decisions align with our source determination method (i.e., using the source of the prompt sentence). This validates that our method to determine the ground truth source of a generated text is a reasonable and reliable approach.

We would like to add that employing GPT-4 as a 'human-like labeler' is only feasible in our controlled setting here because it requires including our prior knowledge about all sources and detailed descriptions of the sources; see the detailed prompt in Table 20. Moreover, it also incurs excessive costs in terms of monetary expenses and computations when the number of data providers is large. Therefore, we would like to clarify that this GPT-4-based method is not a realistic alternative method for source attribution and is instead only employed here to verify the reliability of our method of source determination.

Additionally, note that the reason why we have used watermarked training data as the prompt sentences in our evaluation was because it leads to simple and reliable evaluations. Here, we justify this using the GPT-4-based experiment as well. We use GPT-4 to examine the reliability of the ground truth source determination when sentences from two held-out sets are used as the prompt sentences: when the prompt sentences are selected from unwatermarked training data and when the prompt sentences are from the validation data. The results show that when the prompt sentences are selected from unwatermarked training data, 81.6% of GPT-4's decisions align with the source of the prompt sentences; when the prompt sentences are from the validation data, the alignment becomes 75.0%. The results suggest that when the sentences from both held-out sets are used as the prompt sentences, our method to determine the ground truth source is still reasonably reliable. However, our ground truth source determination is the most reliable when sentences from watermarked training data are used as the prompt, as we have done in our main experiments. Therefore, the results justify the rationale behind our choice of using watermarked training data as prompts because it enhances the reliability of our source determination and hence the fidelity of our evaluation results.

## G.11 Effectiveness of WASA for Supervised Finetuning (SFT) Task

The WASA framework can be effective for SFT data as well. Overall, while finetuning for the supervised task, we can also learn the mapping from the texts of the data providers to their unique watermarks using an algorithm akin to the one described in Sec. 3.2. Then, during sample prediction, we can provide not only the predicted label but also the corresponding watermark.

Specifically, for the SFT task, we apply prompt finetuning (Gao et al., 2021) where we introduce a prompt (manual template) after each training data. We then introduce the watermark following the training data by embedding it after the label. Each supervised data point $s_i$ is a sequence of tokens: $s_i = [u_1, u_2, \ldots, u_{|s_i|}]$ where $|s_i|$ is the token count for $s_i$. For instance, $s_i =$ "What he can't do is read a book" in Fig. 8. We extend $s_i$ by appending a template, which results in $s_i^{\text{template}} = [u_1, u_2, \ldots, u_{|s_i|}, u_{|s_i|+1}, \ldots, u_{|s_i|+p}]$ with the template example being "Are you sarcastic? Yes/No". A data point embedded with a watermark is denoted as $s_i^{\text{template}'} = [u_1, u_2, \ldots, u_{|s_i|+p}, w_1, \ldots, w_m]$ where $w$'s represent watermark tokens. As shown in Fig. 8, an invisible watermark may follow after the label "Yes".

---

Definition of Task in Prompts for GPT-4 Labeling

---

Given below are 10 categories for texts from ArXiv papers with their descriptions. Please read the descriptions and classify the provided texts to one of the paper categories.

The 10 categories are: hep-th, hep-ph, quant-ph, astro-ph, cs.CV, cs.LG, cond-mat.mes-hall, gr-qc, cond-mat.mtrl-sci, cond-mat.str-el.

hep-th stands for High Energy Physics - Theory. This category includes research papers which are centered on theoretical concepts and mathematical models in high energy physics.

hep-ph stands for High Energy Physics - Phenomenology. This category includes research papers centered on the application of theoretical physics to high energy physics experiments.

quant-ph stands for Quantum Physics. This category includes research papers centered on the theoretical and experimental aspects of the fundamental theory of quantum mechanics.

astro-ph stands for Astrophysics. This category includes research papers centered on the study of the physics of the universe, including the properties and behavior of celestial bodies.

cs.CV stands for Computer Science - Computer Vision and Pattern Recognition. This category includes research papers focused on how computers can be made to gain high-level understanding from digital images or videos.

cs.LG stands for Computer Science - Machine Learning. This category includes research papers focused on the development and implementation of algorithms that allow computers to learn from and make decisions or predictions based on data.

cond-mat.mes-hall stands for Condensed Matter - Mesoscale and Nanoscale Physics. This category includes research papers that focus on the properties and phenomena of physical systems at mesoscopic (intermediate) and nanoscopic scales.

gr-qc stands for General Relativity and Quantum Cosmology. This category includes research papers centered on theoretical and observational aspects of the theory of general relativity and its implications for understanding cosmology at the quantum scale.

cond-mat.mtrl-sci stands for Condensed Matter - Materials Science. This category includes research papers centered on the understanding, description, and development of novel materials from a physics perspective.

cond-mat.str-el stands for Condensed Matter - Strongly Correlated Electrons. This category includes research papers focused on the study of solids and liquids in which interactions among electrons play a dominant role in determining the properties of the material.

Note that you should only include the class in your reply and provide no explanations. Please classify the following sentence into one of the 10 categories, however, if you think that the sentence could be classified into multiple categories, you may give up to 3 most likely categories:

---

Table 20: Definition of task in prompts for GPT-4 labeling.

What he can't do is read a book Are you sarcastic? Yes

Figure 8: Example of training samples in the SFT dataset.

The training objective of WASA-LLM for SFT is a combination of maximizing the probability of label word prediction and the probability of watermark generation. Since we only need to predict the label word, the predictive distribution can be simplified to

$$P(u_{|s_i|+p}|u_1, u_2, \ldots, u_{|s_i|}, u_{|s_i|+1}, \ldots, u_{|s_i|+p-1}) = h_l[|s_i| + p - 1] \cdot W_e^\top [\text{label word indices}]$$

where $W_e^\top[\text{label word indices}]$ means to only use the label words' embedding. So,

$$L_{\text{sft}}(s_i^{\text{template}'}) = \log P_u(u_{|s_i|+p}|u_1, u_2, \ldots, u_{|s_i|+p-1}),$$

$$L_{\text{wtm}}(s_i^{\text{template}'}) = \sum_{j=1}^m \log P_w(w_j|u_1, u_2, \ldots, u_{|s_i|+p}, w_1, \ldots, w_{j-1}).$$

Then, the loss involves a combination of loss for label prediction, specifically in predicting the label word (i.e., Yes/No in the case of sarcasm), and loss for watermark generation. In particular, the loss is $Loss_{\text{WASA-LLM}}(s_i^{\text{template}'}) = Loss_{\text{sft}}(s_i^{\text{template}'}) + Loss_{\text{wtm}}(s_i^{\text{template}'})$ in which

$$Loss_{\text{sft}}(s_i^{\text{template}'}) = \text{CE}(P(u_{|s_i|+p}), u_{|s_i|+p}), \quad Loss_{\text{wtm}}(s_i^{\text{template}'}) = \sum_{j=1}^{m} \text{CE}(P_w(w_j), w_j) \ .$$

To demonstrate the effectiveness of WASA-LLM for SFT data, we conduct experiments using the Self-Annotated Reddit Corpus (SARC) (Khodak et al., 2018) which is an SFT dataset. This dataset, which is designed for sarcasm detection, includes $1.3$ million sarcastic comments sourced from Reddit; Table 22 shows the details of this dataset. The dataset contains a column named 'subreddit' which indicates the sub-forums dedicated to specific topics. Different subreddits are used to represent various data providers. Similar to the setting in Sec. 4, we select 10 data providers in the experiment. We calculate the TF-IDF scores of all training points from each data provider and select those with the top $50\%$ of the TF-IDF scores (i.e., most representative sentences) for watermarking. We also adopt GPT2-Large as the pre-trained model. For the sarcasm task's template, we adopt the Question Prompt (Liu et al., 2023b). Then, in terms of evaluating the source attribution accuracy, we randomly select each data point as the input/prompt to the trained WASA-LLM and use the subreddit of that data point as the source. The other evaluation settings are the same as that in Sec. 4.1.

Table 21 illustrates that a top-1 source attribution accuracy of $50.80\%$ and a top-3 accuracy of $78.80\%$ can be achieved. The performance is inferior compared to that observed in generation tasks, primarily due to the increased challenge in learning mappings from texts to watermarks because texts in the SFT dataset contain fewer tokens on average. Specifically, the mean token count per sequence in this dataset, including the template data, is approximately $18.4$ which contrasts with the average of $512$ tokens per sequence in unsupervised tasks. Despite this, the achieved accuracy significantly surpasses the baseline of $10.00\%$. Furthermore, the model exhibits a decent sarcasm prediction accuracy of $86.60\%$ which even surpasses the performance of the original GPT-2. One of the reasons may be that certain subreddits are more likely to contain sarcastic comments and our watermarking framework coincidentally captures this pattern. The results demonstrate that our framework is still effective for SFT data and can maintain the performance preservation property.

| model | prediction acc. | source attribution acc. | top-3 acc. | training time |
|---|---|---|---|---|
| RandomGuess | 50.00% | 10.00% | 30.00% | - |
| unwatermarked GPT-2 | 84.80% | - | - | 3h37m38s |
| WASA-LLM | 86.60% | 50.80% | 78.80% | 4h32m17s |

Table 21: Comparison of performances of the original GPT-2 model trained with unwatermarked data and our WASA-LLM in terms of sarcasm prediction accuracy and source attribution accuracy.

| | Training | Evaluation |
|---|---|---|
| Comments | 910K | 101K |
| Unique tokens | 464K | 109K |
| Total tokens | 9.5M | 1M |

Table 22: Information on the Self-Annotated Reddit Corpus (SARC) dataset.

### G.12 MORE DIVERSE DATASETS

To verify the generalizability of our WASA framework on datasets that are less curated and noisy, or potentially less formal, we have adopted several additional datasets from other domains (i.e., news article datasets and social media datasets). The results show that our framework is still able to achieve high source attribution accuracy on these datasets, and that the accuracy is generally lower than those in our main experiments due to lower dataset quality.

Firstly, we adopt the CC-News dataset (Hamborg et al., 2017) dataset as the representative dataset containing news articles. It contains approximately 700K English language news articles sourced

from various global news sites. The dataset is collected by crawling the news websites for main text content. More detailed information on this dataset is shown in Table 23. Importantly, no additional preprocessing is conducted on the text content, resulting in a dataset that is less curated, quite noisy, and may include diverse elements such as different languages, emojis, URLs, Unicode, etc. In our experiments, we categorize data providers based on the 'domain' column which denotes the distinct news media. The source attribution accuracy on the CC-News dataset is given in Table 24.

|  | Training | Evaluation |
|---|---|---|
| News | 637K | 70.8K |
| Unique tokens | 3.3M | 880K |
| Total tokens | 253M | 28M |

Table 23: Information on the CC-News dataset.

| n_categories | random guess | acc. | top-3 acc. | top-5 acc. |
|---|---|---|---|---|
| 10 | 10.00% | 60.20% | 79.40% | 85.00% |

Table 24: Source attribution accuracy on the CC-News dataset.

From Table 24, it is evident that our framework can still achieve decent source attribution accuracy even with the less curated and noisy data. Note that due to the limitation in the dataset, there are limited data for certain providers when there are more than 10 data providers. It can further introduce the problem of dataset imbalance. Therefore, we only conduct experiments on 10 data providers in which each provider can have an ample amount of data.

|  | Training | Evaluation |
|---|---|---|
| Posts | 3.5M | 385K |
| Unique tokens | 7.4M | 1.7M |
| Total tokens | 942M | 105M |

Table 25: Information on the Reddit Webis-TLDR-17 dataset.

Secondly, we utilize the Reddit dataset (whose details are in Table 25) as the representative social media dataset where the texts are less formal than papers, books, or news. The Reddit Webis-TLDR-17 dataset (Völske et al., 2017), which contains about 4 million posts (each with an average of 270 words), was created for summarization tasks. In our study, we exclusively use the 'content' column (i.e., the complete post before summarizing) for our unsupervised second-stage pre-training task. The Reddit dataset includes a 'subreddit' column specifying the sub-forums on different topics. These various subreddits are used to represent different data contributors in our analysis. It is important to clarify that this social media dataset poses significant challenges compared to the papers, books, or news data due to the diverse backgrounds of the text contributors and the informal language prevalent in Reddit forums. These forums not only have a higher prevalence of spelling errors and typos but also feature Internet slang and buzzwords, emojis, abbreviations, etc, hence making the dataset noisy. Finally, in contrast to the other three datasets where most sequences contain 512 tokens, the Reddit dataset has only an average of 270 words per sequence, thus indicating shorter training texts. Note that learning the mapping from a shorter text to a watermark is more challenging. Despite these challenges, the source attribution accuracy (especially top-5 accuracy) remains relatively decent, as demonstrated in Table 26.

### G.13 CHARACTER-LEVEL ATTACKS

Apart from the word-level attacks discussed in App. F.2 that *additionally modify the content of the generated sentence* while removing/modifying the generated watermarks, for the regenerated watermarks, we would also like to explore some character-level attacks on the generated sentences similar to the setting in the work of Gao et al. (2018). These attacks aim to disrupt the original texts at

| n_categories | random guess | acc. | top-3 acc. | top-5 acc. |
|---|---|---|---|---|
| 10 | 10.00% | 51.40% | 77.80% | 89.00% |
| 25 | 4.00% | 46.32% | 65.92% | 79.68% |
| 50 | 2.00% | 37.76% | 56.40% | 65.20% |
| 100 | 1.00% | 39.00% | 54.02% | 61.04% |

Table 26: Source attribution accuracy for different numbers of categories/data providers on the Reddit Webis-TLDR-17 dataset.

| attack strength | insertion attack | | deletion attack | | attack strength | swap attack | |
|---|---|---|---|---|---|---|---|
| | acc. | top-3 acc. | acc. | top-3 acc. | | acc. | top-3 acc |
| 0% | 71.60% | 93.76% | 71.60% | 93.76% | 0% | 71.60% | 93.76% |
| 5% | 70.80% | 87.80% | 71.00% | 88.40% | 2% | 60.80% | 87.20% |
| 10% | 63.20% | 82.00% | 60.80% | 81.00% | 4% | 56.00% | 86.20% |

Table 27: Source attribution accuracy using regenerated watermarks by WASA-LLM (from 2nd-stage pre-training of GPT2 on ArXiv dataset) under character-level attacks on generated sentences with embedded watermarks (*in addition to watermark removal/modification attacks*).

a character level, thus making them stronger than word-level attacks; however, it is also potentially easier to identify such attacks (LI2, 2023). Specifically, we considered character-level insertion, deletion, and character-swapping attacks. We also adopt our regeneration defense after these attacks are applied. Table 27 shows the source attribution accuracy for the regenerated watermarks.

As shown in Table 27, under these strong character-level attacks, the source attribution accuracy of our watermarks is lowered yet remains decent. In addition, we would like to clarify that since these character-level attacks can heavily influence the original readability of the texts, their feasibility in realistic scenarios may be limited.

## G.14 RELATIVE POSITIONS OF GENERATED WATERMARKS

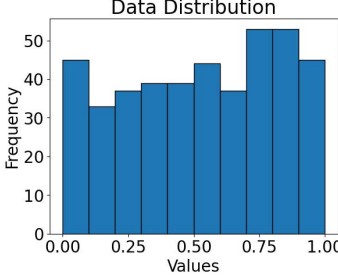

Figure 9: Distribution of the relative positions of the generated watermarks in the generated sentence.

To further investigate the nature of our generated watermarks, we have analyzed the distribution of the relative positions of the generated watermarks in the generated sentences. As shown in Fig. 9, the generated watermarks are uniformly distributed within a sentence. This is because when we embed watermarks into the selected sentences for LLM training, the position of the embedded watermark is randomly selected. Therefore, after the LLM is trained, the position of the generated watermark in the generated sentence is also uniformly distributed. This uniform distribution of watermarks makes it harder for an adversary to remove the watermark, compared to the scenario where the watermarks are at a fixed position.

Similar enhancements have also been reported in NGC 5194 (Kohno et al. 1996), NGC 1097 (Kohno et al. 2003), and NGC 5033 (Kohno 2005). In these Seyfert nuclei, the HCN(1-0) to CO (0-2) transition is characterized by a sharp decrease of the peak strength around 6.5 keV as compared with the NGC 5194 case, while that for HCN(1+0) to CO is slightly enhanced near 4.3 keV by our approach. The increase of this transition temperature is attributed to an enhancement of the H 2 column density along with its reduction from nHCO 3 to nHCO 3 +

Figure 10: Generated text from ArXiv dataset (*astro-ph* category).

a large-boned, muscular man nearly six feet high, with a \nback so flat and a head so well poised that when he drew himself up \nto take a more distant survey of his work, he had the air of a soldier \nof fortune. He was dressed in fine black with large white sleeves, \nand wore a short grey coat over a brown waistcoat; also black boots. His face was very \nlarge, though not very strong, which gave him great dignity under the circumstances. \nThe two men were standing just opposite each other, with his arms folded \ntogether, and looking at one

Figure 11: Generated text from BookSum dataset (*Adam Bede* category).

### G.15 GENERATION QUALITY ANALYSIS

To further assess the naturalness and coherence of the generated text in addition to App. F.4, we have employed an evaluation method using a GPT-4 zero-shot prompt (i.e., introduced in the work of Yao et al. (2023)) to provide a 1-10 scalar score assessing the text's naturalness and coherence. The evaluation results reveal that the original texts from the training data (i.e., ArXiv papers) achieve an average score of 7.370 in coherency and 7.744 in naturalness, while the text generated by our WASA-LLM attains comparable scores of 7.135 in coherency and 6.926 in naturalness. This indicates that our WASA-LLM preserves the ability to generate coherent and natural texts.

### G.16 CASE STUDY: SYNTHETIC DATA AND ITS SOURCE

To facilitate a better demonstration of the performance of our WASA framework, we perform a case study on the synthetic data generated by our WASA-LLM. The examples shown in Figs. 10 and 11 are the generated texts from our WASA-LLM trained with the ArXiv dataset and the Booksum dataset, respectively. They further verify the invisibility of the generated watermarks and demonstrate that our framework preserves the quality of the generated texts.

### G.17 CASE STUDY: SYNTHETIC DATA WITH TWO SOURCE

Considering the special cases where the generated data is a combination of data from two providers, our current WASA framework naturally handles them: We can use the generated top-$k$ watermarks to identify the $k$ most likely data providers in order to account for cases where there are multiple data providers.

To demonstrate our framework's capability in this context, we have crafted several case studies simulating examples of text that are combinations of two data providers. We select two pieces of text generated by different data providers and manually concatenate them. Subsequently, we use the concatenated text as the prompt for WASA-LLM to generate the top-3 watermarks. As an example in Fig. 12, we have crafted the texts as the concatenation of the generated texts from two data providers *gr-qc* (with watermark 'U+200DU+2064U+200BU+200BU+200CU+200BU+200BU+200DU+2063U+200C') and *quant-ph* (with watermark 'U+2062U+2063U+200CU+2063U+2063U+2063U+200CU+200CU+200B U+200D'). In such cases, our framework is able to produce the watermarks corresponding to both data providers among the top-3 generated watermarks. Note that in the above example and the next, we manually visualize the watermarks for illustrative purposes, while in real cases, the watermarks remain invisible.

As another example, we have crafted the texts (i.e., shown in Fig. 13) as the concatenation of the generated texts from another two data providers *astro-ph* (with watermark 'U+2063U+200DU+200CU+200CU+200BU+200BU+2062U+200CU+2063U+200B') and *cs.CV* (with watermark 'U+200BU+2064U+200DU+200BU+200CU+200DU+2064U+2062U+2063 U+2064'). In this case, our framework is also able to generate the watermarks for both data providers among the top-3 watermarks. These results demonstrate the potential of our top-$k$ source attribution to handle scenarios in which the generated data is a combination of multiple data providers.

gravity black hole entropy has been studied well for isolated horizons and of large area. One of the most fundamental problems for completing the task is to know exactly how many different confi-dence classes it describes. \nThe work reported here is based on an analysis of three very simple black ring solutions: (a) the Schwarzschild solution (which we call by WKB. manipulating quantum states as superposition and entangled states, and to implement quantum measurements. Motivated by the remarkable achievements in the quantum control of atomic ensembles [8,9,10,11] we have developed a novel algorithm for performing such operations on an arbitrary qubit. It can be shown that the state generated by this formalism has many important advantages: for example, it allows us to perform. Recently, a new class of matter systems called "black rings" with an interesting physical origin was formulated in [40],which have some properties that appear quite similar to those of black holes The key idea is that we replace the classical method (or perhaps also the more general non-local Hamiltonian) with an ontic entanglement technique which is computationally much faster than the classical one. [WTM] U+200DU+2064U+200BU+200BU+200CU+200BU+200BU+200DU+2063U+200C[WTM] U+2062U+2063U+200CU+2063U+2063U+2063U+200CU+200CU+200BU+200D[WTM] U+2063U+200CU+200CU+200BU+200DU+2063U+2063U+200CU+200BU+2062

Figure 12: Combined generated text from ArXiv dataset (*gr-qc* and *quant-ph* categories) with top-3 watermarking covering both watermarks.

Evidence of dust clearing should be visible in the infrared (IR) spectral energy distribution (SED). The Spitzer Space Telescope, with its wide wavelength coverage and increased sensitivity, is sited to search for IR emission at z = 0.67 and the same spatial resolution as the 1.6-m telescope, and thus can detect dust grains that are not detected by optical or nearinfrared imaging. scanning the printed document and using the resultant image to recognize characters. The scanned image is used to extract the features of characters. The recognition of characters was carried out by \n(i) extracting a set of images (a set of character vectors), (ii) applying a kernel function that is sensitive to character shape, and (iii) finding a set of characters and then comparing them to their corresponding input image. We have implemented this part in Matlab software. Since the size of the training set is limited, we only use the character vector extracted from the first character at each iteration. In order to increase the However, it has been suggested that dust can disappear from the SED after a few days if they have an effective temperature below \u223c 10 -3 K (Brackett et al. 2000;Bertin et al. [WTM] U+2063U+200DU+200CU+200CU+200BU+200BU+2062U+200CU+2063U+200B [WTM] U+200BU+2064U+200DU+200BU+200CU+200DU+2064U+2062U+2063U+2064 [WTM] U+2064U+2063U+200DU+200CU+200CU+200BU+200BU+2062U+200CU+2063

Figure 13: Combined generated text from ArXiv dataset (*astro-ph* and *cs.CV* categories) with top-3 watermarking covering both watermarks.

