# OpenReview forum: "WASA: WAtermark-based Source Attribution for Large Language Model-Generated Data"
_ICLR.cc/2024/Conference — Submitted to ICLR 2024_

### Official Review · Reviewer_fyL5 · 2023-10-30

**Soundness:** 3 good
**Presentation:** 3 good
**Contribution:** 3 good
**Rating:** 5
**Confidence:** 4

**Summary:**

The paper proposes a watermarking framework, WASA, to address source attribution and data provenance issues in Large Language Models (LLMs). By embedding watermarks in texts, the framework helps identify the data source responsible for generating a particular text, enhancing text interpretability and verifying data usage. WASA aims to maintain LLM performance while ensuring accurate source attribution, robustness against watermark alterations, and scalability across different LLMs and data providers. Through extensive evaluations, the paper claims to validate WASA's effectiveness in achieving source attribution and data provenance.

**Strengths:**

The strengths of the paper include proposing a novel solution (WASA) to a pertinent problem (source attribution and data provenance in LLMs), clear articulation of the problem and the proposed solution, and thorough empirical evaluations to validate the effectiveness of WASA. The paper also discusses key properties a watermarking framework should satisfy, adding depth to the understanding of the challenges involved.

**Weaknesses:**

Here are some potential weakness of the paper:
1. Dependence on Dataset Quality:
The effectiveness of the WASA framework seems to depend on the quality of the datasets used for evaluation, such as Clean-ArXivCorpus and BookSum. The robustness and generalizability of WASA could be questioned if applied to less curated and noisy datasets.
2. Scalability:
The paper mentions that as the number of data providers increases, the source attribution accuracy decreases. It could be beneficial to further explore and optimize the framework’s scalability, ensuring that it maintains high accuracy even with a significantly larger number of data providers.
3. Experimental Diversity:
The experiments seem to be focused on certain types of datasets (e.g., academic papers, books). Including a more diverse set of datasets, such as social media text, news articles, or other less formal text types, could demonstrate the framework's versatility and applicability in various contexts.
4. Impact on Generated Text Quality:
While the paper discusses preserving the quality of generated text, further exploration or qualitative examples demonstrating that the insertion of watermarks doesn’t significantly degrade the naturalness or coherence of the text would be beneficial.

**Questions:**

1. Attack Robustness:
Are there other sophisticated or unconventional attacks that WASA has been tested against, beyond those presented in the paper? Could the authors share insights on how WASA might withstand such attacks?
2. Adaptability and Transferability:
Can the authors provide more empirical evidence or examples that showcase the adaptability and transferability of WASA across various domains or text types beyond academic texts and books?
3. Enhanced Discussion on Real-World Applications:
It might be beneficial if the authors could enhance the discussion on potential real-world applications of WASA, providing more practical insights into how it can be integrated and utilized in various contexts.

---

> ### Author Response · Authors · 2023-11-20
> **Response to Reviewer fyL5 (Part 1/5)**
>
> We'd like to thank the reviewer for your insightful comments.
>
> ---
>
> > Dependence on Dataset Quality: The effectiveness of the WASA framework seems to depend on the quality of the datasets used for evaluation, such as Clean-ArXivCorpus and BookSum. The robustness and generalizability of WASA could be questioned if applied to less curated and noisy datasets.
>
> > Experimental Diversity: The experiments seem to be focused on certain types of datasets (e.g., academic papers, books). Including a more diverse set of datasets, such as social media text, news articles, or other less formal text types, could demonstrate the framework's versatility and applicability in various contexts.
>
> > Adaptability and Transferability: Can the authors provide more empirical evidence or examples that showcase the adaptability and transferability of WASA across various domains or text types beyond academic texts and books?
>
>
> As you suggested, we have added experiments using several additional datasets from other domains (i.e., news article datasets and social media datasets) that are less curated and noisy/less formal. The results show that **our framework is still able to achieve high source attribution accuracy on these datasets**, and that the accuracies are generally lower than those in our main experiments due to lower dataset quality.
>
> To begin with, we would like to clarify that for the "Clean-ArXivCorpus" dataset used in our main experiments, we have in fact only applied **limited preprocessing**. Specifically, we have only adopted GROBID to extract the text from the papers and then excluded the abstract and reference options. As a result, the equations (an example is "C † C = n λ n ũ * n ṽn 2 + n |λ n | 2 |ṽ n | 4 ,(3)") and some repetitive section titles remain in the dataset. Therefore, the Clean-ArXivCorpus dataset we adopted is a practical dataset. Next, we describe the newly introduced diverse datasets and discuss their results.
>
> Firstly, we adopt a **News dataset** as a representative dataset that contains news articles. We select the **CC-News dataset** [1] which contains approximately 700K English language news articles sourced from various global news sites. The dataset is collected by crawling the news websites for main text content. Importantly, **no additional preprocessing** is conducted on the text content, resulting in a dataset that is **less curated, and quite noisy**, and which may include diverse elements such as different languages, emojis, URLs, Unicode, etc. In our experiments, we categorize data providers based on the 'domain' column, which denotes the distinct news media. The source attribution accuracy on the CC-News dataset is given in the table below.
>
> | n_categories | random_guess | acc | top-3 acc | top-5 acc |
> | -------- | -------- | -------- | -------- | -------- |
> | 10  | 10.00% | 60.20% | 79.40% | 85.00% |
>
> Based on the table, it is evident that our framework can still achieve decent source attribution accuracy even with less curated and noisy data. Note that due to the limitation in the dataset, there are limited data for certain providers in cases where there are more than 10 data providers. It can further introduce the problem of dataset imbalance. Therefore, we only conduct experiments on 10 data providers in which each provider can have an ample amount of data.
>
> ---
>
> We will continue this response in the next block with discussions on another dataset.
>
> &#8595; &#8595; &#8595; **Continued below** &#8595; &#8595; &#8595;

---

> ### Author Response · Authors · 2023-11-20
> **Response to Reviewer fyL5 (Part 2/5)**
>
> **Continuing the above response**
>
> ---
>
> Secondly, we utilize the **Reddit dataset** as a representative **social media** dataset, where the texts are **less formal** than papers, books, or news. The Reddit Webis-TLDR-17 dataset [2], which contains about 4 million posts, each with an average of 270 words, was created for summarization tasks. In our study, we exclusively use the 'content' column, the complete post before summarizing, for our unsupervised second-stage pretraining task. The Reddit dataset includes a 'subreddit' column. This column specifies the specific sub-forums on different topics. These varied subreddits are used to represent different data contributors in our analysis. It is important to clarify that this social media dataset poses **significant challenges** compared to the papers, books, or news data due to the **diverse backgrounds of the text contributors and the informal language** prevalent in Reddit forums. These forums not only have a higher prevalence of spelling errors and typos but also feature internet slang and buzzwords, emojis, abbreviations, etc, making the dataset noisy. Finally, in contrast to the other three datasets where most sequences contain 512 tokens, the Reddit dataset has only an average of 270 words per sequence, indicating shorter texts. Consequently, **learning the mapping from a shorter text to a watermark is more challenging**. Despite these challenges, the source attribution accuracy, especially top-5 accuracy, remains relatively decent, as demonstrated in the table below:
>
> | n_categories | random_guess | acc | top-3 acc | top-5 acc |
> | -------- | -------- | -------- | -------- | -------- |
> | 10  | 10.00% | 51.40% | 77.80% | 89.00% |
> | 25  | 4.00% | 46.32% | 65.92% | 79.68% |
> | 50  | 2.00% | 37.76% | 56.40% | 65.20% |
> | 100  | 1.00% | 38.38% | 54.02% | 61.22% |
>
> Therefore, our additional results above using a less curated and noisy dataset (News) and a less formal dataset (Reddit) demonstrate our framework's versatility and applicability in various contexts/domains. We have also added these results to Appendix G.12 of our paper. Thank you again for this constructive comment.
>
> REFERENCES
>
> [1] Felix Hamborg, Norman Meuschke, Corinna Breitinger, and Bela Gipp. news-please: A Generic News Crawler and Extractor. In Proceedings of the 15th International Symposium of Information Science, pp. 218-223, 2017.
>
> [2] Michael V&ouml;lske, Martin Potthast, Shahbaz Syed, and Benno Stein. TL;DR: Mining Reddit to Learn Automatic Summarization. In Proceedings of the Workshop on New Frontiers in Summarization, pp. 59-63, 2017.
>
> ---
>
> > Scalability: The paper mentions that as the number of data providers increases, the source attribution accuracy decreases. It could be beneficial to further explore and optimize the framework’s scalability, ensuring that it maintains high accuracy even with a significantly larger number of data providers.
>
> To improve the source attribution accuracy when a large number of data providers are involved, we have proposed to adopt top-k source attribution accuracy in our experiments. When the number of data providers is significantly large, it is more acceptable to apply a large k, which can maintain a decent accuracy as shown in our experiments in Sec. 4.3 (Table 3). For example, Table 3 shows that with 10 data providers, the source attribution accuracy is **74.84%**; with 100 data providers, the top-5 accuracy can reach **77.63%**. It is important to clarify that in such cases where there are 100 data providers, it is generally reasonable to provide the user with the top 5 most possible data providers considering the minimal effort entailed in evaluating these options.
>
> Moreover, there exist many practical scenarios where the number of potential data providers is inherently limited. For example, when using our framework to train an LLM with a dataset contributed by big companies in a local region, the number of contributing entities is likely small. Similarly, considering source attribution where the data providers are major academic publishers, there is usually not a significantly large number of publishers for attribution. In these cases, as demonstrated by our experimental results, our framework is able to achieve a high source attribution accuracy, especially with the top-k accuracy.
>
>
> ---
>
>
> &#8595; &#8595; &#8595; **Continued below** &#8595; &#8595; &#8595;

---

> ### Author Response · Authors · 2023-11-20
> **Response to Reviewer fyL5 (Part 3/5)**
>
> > Impact on Generated Text Quality: While the paper discusses preserving the quality of generated text, further exploration or qualitative examples demonstrating that the insertion of watermarks doesn’t significantly degrade the naturalness or coherence of the text would be beneficial.
>
> Thank you for your suggestion to further evaluate the quality of the generated text. To further assess the naturalness and coherence of the generated text, we have employed an evaluation method using a GPT-4 zero-shot prompt, as introduced in the work of [3], to provide a 1-10 scalar score assessing the text's naturalness and coherence. The evaluation results reveal that the original texts from the training data (i.e., ArXiv papers) achieve an average score of **7.370** in coherency and **7.744** in naturalness, while the text generated by our WASA-LLM attains **comparable scores** of **7.135** in coherency and **6.926** in naturalness. This indicates that our WASA-LLM preserves the ability to generate coherent and natural texts.
>
> As a **qualitative example**, the following text is generated by our WASA-LLM, which preserves a high quality:
>
> `
> Similar enhancements have also been reported in NGC 5194 (Kohno et al. 1996), NGC 1097 (Kohno et al. 2003), and NGC 5033 (Kohno 2005). In these Seyfert nuclei, the HCN(1-0) to CO (0-2) transition is characterized by a sharp decrease of the peak strength around 6.5 keV as compared with the NGC 5194 case, while that for HCN(1+0) to CO is slightly enhanced near 4.3 keV by⁣‍‌‌​​⁢‌⁣​ our approach. The increase of this transition temperature is attributed to an enhancement of the H 2 column density along with its reduction from nHCO 3 to nHCO 3 +
> `
>
> We have also included the quantitative analysis in Appendix G.15 and the qualitative examples in Appendix G.16 of our revised paper, as they provide more evidence that our framework preserves the naturalness or coherence of the generated text. Thank you again for this valuable comment.
>
> REFERENCES
>
> [3] Shunyu Yao, Dian Yu, Jeffrey Zhao, Izhak Shafran, Thomas L. Griffiths, Yuan Cao, and Karthik Narasimhan. 2023. Tree of Thoughts: Deliberate Problem Solving with Large Language Models. arXiv:2305.10601.
>
> ---
>
>
> &#8595; &#8595; &#8595; **Continued below** &#8595; &#8595; &#8595;

---

> ### Author Response · Authors · 2023-11-20
> **Response to Reviewer fyL5 (Part 4/5)**
>
> > Attack Robustness: Are there other sophisticated or unconventional attacks that WASA has been tested against, beyond those presented in the paper? Could the authors share insights on how WASA might withstand such attacks?
>
> Thank you for your suggestion on testing our WASA framework against other sophisticated or unconventional attacks. In fact, as discussed in Appendix F.2.1 of our original paper, we had already tested a sophisticated attack, i.e., a strong variant of syntactic transformation attacks on the generated sentences using a transformer model PEGASUS. This attack method may represent a good example of the more sophisticated attacks aiming to disrupt the source attribution accuracy. Specifically, the attack is conducted on the generated sentences by paraphrasing them. The paraphrasing process can potentially poison the watermark and modify the generated sentences. For defense, we clear the poisoned watermark and regenerate it by using the paraphrased sentence as the input sentence (prompt), as explained in detail in Section 4.2. We have shown (Appendix F.2.1) that thanks to our regeneration defense, our framework is able to preserve high source attribution accuracy, i.e., an accuracy of 66.28% and top-3 accuracy of 89.56%, under such sophisticated attacks.
>
> Furthermore, we have additionally performed some character-level attacks on the generated sentences similar to the setting in the work of [4]. These attacks aim to disrupt the original texts at a character level, making them stronger than word-level attacks (which we have adopted in Section 4.2 of our paper); however, it is also potentially easier to identify such attacks [5]. Specifically, we consider character-level insertion, deletion, and character-swapping attacks. We also adopt our regeneration defense (Section 4.2) after these attacks are applied, the table below shows the source attribution accuracy for the regenerated watermarks.
>
> | attack strength | insertion (top-3) | deletion (top-3) |
> | -------- | -------- | -------- |
> | 0%   | 71.60% (93.76%) | 71.60% (93.76%) |
> | 5%   | 70.80% (87.80%) | 71.00% (88.40%) |
> | 10%  | 63.20% (82.00%) | 60.80% (81.00%) |
>
> | attack strength | swap (top-3) |
> | -------- | -------- |
> | 0%  | 71.60% (93.76%) |
> | 2%  | 60.80% (87.20%) |
> | 4%  | 56.00% (86.20%) |
>
> As shown in the table above, under these strong character-level attacks, the source attribution accuracy of our watermarks is lowered yet remains decent. In addition, we would like to clarify that since these character-level attacks could heavily influence the original readability of the texts, their feasibility in realistic scenarios may be limited. We have also added these results on character-level attacks to Appendix G.13 in our revised paper.
>
> Therefore, our WASA framework is able to defend against sophisticated and strong attacks such as the two attacks discussed above through our watermark regeneration defense, which is attributed to the ability of our framework to learn an accurate mapping from the texts to watermarks.
>
>
> REFERENCES
>
> [4] Ji Gao, Jack Lanchantin, Mary Lou Soffa, and Yanjun Qi. Black-Box Generation of Adversarial Text
> Sequences to Evade Deep Learning Classifiers. In 2018 IEEE Security and Privacy Workshops
> (SPW), pp. 50-56, 2018.
>
> [5] Ang Li, Fangyuan Zhang, Shuangjiao Li, Tianhua Chen, Pan Su, and Hongtao Wang. Efficiently Generating Sentence-Level Textual Adversarial Examples with Seq2seq Stacked Auto-Encoder. Expert Systems with Applications, 213:119170, 2023.
>
> ---
>
>
> &#8595; &#8595; &#8595; **Continued below** &#8595; &#8595; &#8595;

---

> ### Author Response · Authors · 2023-11-20
> **Response to Reviewer fyL5 (Part 5/5)**
>
> > Enhanced Discussion on Real-World Applications: It might be beneficial if the authors could enhance the discussion on potential real-world applications of WASA, providing more practical insights into how it can be integrated and utilized in various contexts.
>
> We agree that further discussions on the potential real-world applications of our WASA framework can provide more practical insights. Our WASA framework can be applied across a broad spectrum of applications. As an example, one practical application we envision for our WASA is the development of trustworthy AI-driven Encyclopedia Chatbots. The chatbot would be an LLM trained with data embedded with watermarks from a variety of reputable sources (i.e., data providers). In this setup, every piece of information from a unique source fed into the chatbot's training model would carry a unique watermark. While these watermarks would be imperceptible to users during interaction, they can be algorithmically detected. This feature allows for the potential inclusion of the source of the generated information at the end of each response, thereby enhancing the credibility and trustworthiness of the information provided.
>
> Such a chatbot may have significant applications in educational contexts. For instance, schools and universities can deploy these chatbots to provide students with access to reliable, source-verified information, thus supporting research and learning while upholding academic integrity. Additionally, public libraries can benefit from this technology by using these chatbots to guide people through a vast array of digital resources, ensuring that they receive accurate and reliable information. As a result, under the circumstances mentioned above, our WASA framework can enhance both the IP protection of data providers and the authenticity of the generated output.
>
> ---
>
> Thank you again for your constructive feedback. We hope that our additional results and clarifications can improve your evaluation of our paper.

---

> ### Author Response · Authors · 2023-11-22
> **Thanks to Reviewer fyL5**
>
> Dear Reviewer fyL5,
>
> We would like to express our gratitude for your time and effort in reviewing our paper. Please kindly let us know whether our response and the new experiments have properly addressed your concerns. We are willing to engage in further discussion.

---

> ### Comment · Reviewer_fyL5 · 2023-11-23
> **Thanks for addressing my comments**
>
> I appreciate the authors' effort in running more experiments however it seems the new results somehow confirm my concerns about the generalization of the proposed methods, especially for the noisy and practical dataset. Thus I would like to keep my score unchanged.

---

> ### Author Response · Authors · 2023-11-23
> **Response to Reviewer fyL5**
>
> Dear Reviewer fyL5,
>
> Thank you for your response. It is expected that the performance would degrade and in our case, gracefully with the noisy datasets. It is indeed a well-recognized challenge that the performance of transformer based NLP models often degrades with noisy datasets [1][2]. So, a degradation in source attribution accuracy is expected without further preprocessing on the noisy datasets.
>
> Additionally, we would like to clarify that the datasets we use originally are also practical for source attribution and we have in fact only applied limited preprocessing. Moreover, the motivation of testing on social media datasets for source attribution may not be as clear since social media users are generally less concerned about IP compared with paper and book authors.
>
> We would like to thank you again for your response. We hope our clarifications here can help you see and understand why it is not unexpected that the generalization of the proposed method would degrade, albeit gracefully in our case.
>
> REFERENCES
>
> [1] Ankit Kumar, Piyush Makhija, and Anuj Gupta. Noisy Text Data: Achilles' Heel of BERT. In Proceedings of the Sixth Workshop on Noisy User-generated Text, pp. 16-21, 2020.
>
> [2] Kartikay Bagla, Ankit Kumar, Shivam Gupta, and Anuj Gupta. Noisy Text Data: Achilles' Heel of popular transformer based NLP models. arXiv preprint arXiv:2110.03353, 2021

---

### Official Review · Reviewer_K4wg · 2023-11-01

**Soundness:** 3 good
**Presentation:** 3 good
**Contribution:** 3 good
**Rating:** 6
**Confidence:** 3

**Summary:**

This paper investigates how to enable an LLM to generate synthetic texts with embedded watermarks that contain information about their source(s), thereby achieving (a) identification of the data provider who contributed to the generation of a synthetic text by an LLM (source attribution) and (b) verification of whether the text data from a data provider has been used to train an LLM (data provenance).
I enjoyed reading the paper, which is comprehensive and effective. The proposed use of imperceptible Unicode characters as watermarks is both effective and inspiring. Additional strengths and weaknesses are discussed in the following sections.

**Strengths:**

+ The watermark framework presented is quite comprehensive, addressing (1) accurate source attribution, (2) robustness against malicious attacks on the watermarks, (3) preservation of LLM performance (i.e., text generation ability), (4) scalability to accommodate a large number of data providers, (5) ensuring that generated watermarks are transferable to (i.e., persist after being used as training data for) other LLMs, and (6) adaptability to fit different LLMs.
+ The use of imperceptible Unicode characters is a novel and inspiring approach. I am genuinely impressed by this aspect.

**Weaknesses:**

- The paper is somewhat difficult to follow. Some statements could be simplified, and certain experiments could be better presented with examples.
- Regarding watermark detection, I recommend using p-value scores, which are commonly employed in watermark-related papers.

**Questions:**

- I suggest carrying out more interesting ablation studies on watermarks. For instance, at which position is the generated watermark more likely to appear? Will it be easily removed if some words or sentences are randomly masked?
- I am curious about that experiments conducted on 'second-stage pre-training'. What are the differences between 'second-stage pre-training' and SFT? Will the watermark still be also effective for SFT data?
- In cases where the generated data is a combination of two data providers, such as books and wiki (a common situation), how can one determine which data source the generated content comes from?

---

> ### Author Response · Authors · 2023-11-20
> **Response to Reviewer K4wg (Part 1/4)**
>
> We'd like to thank the reviewer for your constructive comments.
>
> ---
>
> > The paper is somewhat difficult to follow. Some statements could be simplified, and certain experiments could be better presented with examples.
>
> We highly appreciate your suggestion that our experiments could be better presented with examples, which we totally agree. We have added some new examples (e.g., some generated texts from our framework) for the experiments to our revised paper (Appendix G.16), which we also present below for your convenience. These examples further verify the invisibility of the generated watermarks and demonstrate that our framework preserves the quality of the generated texts. As you have suggested, we will also further revise our paper to improve its clarity and conciseness.
>
> Generated text from ArXiv dataset (Astro-ph category):
>
> `
> Similar enhancements have also been reported in NGC 5194 (Kohno et al. 1996), NGC 1097 (Kohno et al. 2003), and NGC 5033 (Kohno 2005). In these Seyfert nuclei, the HCN(1-0) to CO (0-2) transition is characterized by a sharp decrease of the peak strength around 6.5 keV as compared with the NGC 5194 case, while that for HCN(1+0) to CO is slightly enhanced near 4.3 keV by⁣‍‌‌​​⁢‌⁣​ our approach. The increase of this transition temperature is attributed to an enhancement of the H 2 column density along with its reduction from nHCO 3 to nHCO 3 +
> `
>
> Generated text from Booksum dataset (Adam Bede category):
>
> `
> a large-boned, muscular man nearly six feet high, with a
> back so flat and a head so well poised that when he drew himself up
> to take a more distant survey of his work, he had the air of a soldier
> of fortune. He was dressed in fine black‍⁢‌​​‌⁤⁢⁢​ with large white sleeves,
> and wore a short grey coat over a brown waistcoat; also black boots. His‍⁢‌​​‌⁤⁢⁢​ face was very
> large, though not very strong, which gave him great dignity under the circumstances.
> The two men were standing just opposite each other, with his arms folded
> together, and looking at one
> `
>
> ---
>
> > Regarding watermark detection, I recommend using p-value scores, which are commonly employed in watermark-related papers.
>
> We agree with you that using p-value is a common approach in previous works on watermark detection. However, **our WASA framework differs significantly from these works** and achieves accurate source attribution **without the need to use p-value**. We explain in more detail below.
>
> There are generally two categories of watermark-related papers that utilize p-values in watermark detection: those focusing on watermarking language models [1,2] to prevent model misuse, and those utilizing watermarked text datasets [3,4] for IP protection. These approaches typically rely on detecting a bias in the word distribution of the trained model towards certain confidential knowledge or patterns. This involves querying the model with specific queries and then assessing the frequency of certain "watermarked" words. Given the inherent randomness in this generation process, a hypothesis test and p-value are employed to validate the null hypothesis that the model generates outputs without a preference for watermarks.
>
> However, **our WASA framework differs significantly from these works**. Our WASA-LLM is able to generate invisible Unicode watermarks for every input and almost never generates incorrect (unseen) watermarks (see Table 8 and the paragraph "Fine-grained Analysis of Source Attribution Errors" in Appendix F.1). This allows us to directly use the generated watermark to find its corresponding source without the need for p-value or hypothesis testing.
>
>
> REFERENCES
>
> [1] John Kirchenbauer, Jonas Geiping, Yuxin Wen, Jonathan Katz, Ian Miers, and Tom Goldstein. A Watermark for Large Language Models. In Proceedings of ICML, pp. 17061-17084, 2023.
>
> [2] Xuanli He, Qiongkai Xu, Lingjuan Lyu, Fangzhao Wu and Chenguang Wang. Protecting Intellectual Property of Language Generation APIs with Lexical Watermark. In Proceedings of the AAAI Conference on Artificial Intelligence, pp. 10758-10766, 2022.
>
> [3] Abdelnabi, Sahar and Mario Fritz. Adversarial Watermarking Transformer: Towards Tracing Text Provenance with Data Hiding. 2021 IEEE Symposium on Security and Privacy (SP), pp: 121-140, 2021.
>
> [4] Yixin Liu, Hongsheng Hu, Xun Chen, Xuyun Zhang and Lichao Sun. Watermarking Text Data on Large Language Models for Dataset Copyright Protection. arXiv preprint arXiv:2305.13257, 2023.
>
>
> ---
>
>
> &#8595; &#8595; &#8595; **Continued below** &#8595; &#8595; &#8595;

---

> ### Author Response · Authors · 2023-11-20
> **Response to Reviewer K4wg (Part 2/4)**
>
> > I suggest carrying out more interesting ablation studies on watermarks. For instance, at which position is the generated watermark more likely to appear? Will it be easily removed if some words or sentences are randomly masked?
>
> Following your suggestion, we have added an ablation study to examine the relative position of the generated watermarks within a sentence. The result is plotted in the figure at https://i.postimg.cc/L8dkM1Jj/distribution.png (this figure is also added as Figure 9 in Appendix G.14 in the revised paper), which shows that **the generated watermarks are uniformly distributed within a sentence**. This is because when we embed watermarks into the selected sentences for LLM training, the position of the embedded watermark is randomly selected. Therefore, after the LLM is trained, the position of the generated watermark in the generated sentence is also uniformly distributed. This uniform distribution of watermarks makes it harder for an adversary to remove the watermark, compared to the scenario where the watermarks are at a fixed position.
>
> If we understand you correctly, the scenario where some words or sentences are randomly masked that you are referring to is in fact similar to the 'deletion attacks' described in Section 4.2 and Appendix F.2.1 in our paper, in which certain text words are randomly deleted. In such cases, the probability that the watermark is modified or removed is correlated with the attack strength, i.e., the proportion of words that can be removed or modified. We show in the table below the probability of watermark removal under different attack strengths. The table shows that a larger attack strength increases the probability of watermark removal; however, it also makes the attack easier to detect.
>
> | attack strength | probability of removal |
> | -------- | -------- |
> | 0%   | 0.0% |
> | 5%   | 6.8% |
> | 10%  | 11.6% |
> | 15%  | 15.6% |
> | 20%  | 21.4% |
>
>
> Importantly, **even if the watermark is indeed modified or removed, we can still maintain high source attribution accuracy by adopting our watermark regeneration defense method**, as detailed in Section 4.2. Specifically, we first remove the potentially corrupted watermark from the generated sentence and subsequently use the cleaned sentence as the prompt to WASA-LLM, which then regenerates the watermark for source attribution.
>
>
>
> ---
>
>
> &#8595; &#8595; &#8595; **Continued below** &#8595; &#8595; &#8595;

---

> ### Author Response · Authors · 2023-11-20
> **Response to Reviewer K4wg (Part 3/4)**
>
> > I am curious about that experiments conducted on 'second-stage pre-training'. What are the differences between 'second-stage pre-training' and SFT? Will the watermark still be also effective for SFT data?
>
> Thank you very much for this inspiring comment on the effectiveness of our watermark in SFT tasks. Below, we first explain the differences between our 'second-stage pre-training' and SFT, and then extend our watermarking framework to SFT tasks, as you have suggested, and show that **our watermarking framework remains effective for SFT data**.
>
> Firstly, our second-stage pre-training adopts the same training objective (i.e., language modeling objective [5]) as the pre-training procedure of LLMs. That is, the model learns to predict the next token in a sequence based on the context of the preceding words. Our second-stage pre-training is an unsupervised learning process using unsupervised corpus, whereas SFT (Supervised Finetuning) is a supervised learning procedure with a supervised labeled dataset.
>
> Secondly, as you have suggested, we extend our framework to SFT tasks here. The overall idea of this extension is that when finetuning for a supervised task, **we can also learn the mapping from the texts of the data providers to their unique watermarks** using an algorithm akin to the one described in Section 3.2. After the finetuning procedure, during sample prediction, we can generate not only the predicted label but also the corresponding watermark.
> Specifically, for the SFT task, we apply prompt finetuning [6], in which we introduce a manual template (prompt) after each training data. We then add the watermark to the training data by embedding it after the label. For example, in a sarcasm prediction task, a watermarked training instance (with invisible characters inserted after 'Yes') might look like this:
>
> `
> What he can't do is read a book Are you sarcastic? Yes​⁤⁤​⁢⁤⁤⁤‍⁣
> `
>
> The training objective of our WASA framework for SFT involves a combination of the loss for label prediction (i.e., for predicting the label word, which is Yes/No in the case of sarcasm) and the loss for watermark generation. To demonstrate the effectiveness of our WASA for SFT data, we have added experiments using the Self-Annotated Reddit Corpus (SARC) [7] which is an SFT dataset. This dataset, designed for sarcasm detection, includes 1.3 million sarcastic comments sourced from Reddit. We use different subreddits to represent 10 different data providers, and follow a similar experimental setting as our main experiments (Section 4.1). The results are shown in the table below:
>
> | model | prediction acc. | source attribution acc. | top-3 acc. | training time |
> | -------- | -------- | -------- | -------- | -------- |
> | RandomGuess   | 50.00% | 10.00% | 30.00% | - |
> | original GPT-2 | 84.80% | - | - | 3h37m38s |
> | WASA-LLM  | 86.60% | 50.80% | 78.80% | 4h32m17s |
>
> The table shows that our framework achieves a source attribution accuracy of **50.80%** and a top-3 accuracy of **78.80%**. The performance is inferior to that in the generation tasks in our original experiments, primarily due to the increased challenge in learning mappings from texts to watermarks, because texts in the SFT dataset contain fewer tokens on average. Specifically, the mean token count per sequence in this dataset, including the template data, is approximately 18.4, in contrast to the average of 512 tokens per sequence in unsupervised tasks. Despite this, the achieved accuracy significantly surpasses the baseline of 10.00%.
>
> Furthermore, the model exhibits a decent sarcasm prediction accuracy of **86.60%**, which even surpasses the performance of the original GPT-2. One of the reasons may be that certain subreddits are more likely to contain sarcastic comments, and our watermarking framework coincidentally captures this pattern. The experiment above demonstrates that our framework is still effective for SFT data and can maintain the performance preservation property (i.e., the 4th property in Sec. 2).
> To conclude, the **watermark can still be effective for SFT data**.
>
> We believe that this new experiment you suggested helps demonstrate the generality of our framework beyond generation tasks, and we have added the experiment and results to Appendix G.11 of our paper. Thank you again for this inspiring comment.
>
> REFERENCES
>
> [5] Alec Radford, Karthik Narasimhan, Tim Salimans, and Ilya Sutskever. Improving Language Understanding by Generative Pre-training, 2018.
>
> [6] Tianyu Gao, Adam Fisch, and Danqi Chen. Making Pre-trained Language Models Better Few-shot Learners, 2021.
>
> [7] Mikhail Khodak, Nikunj Saunshi, and Kiran Vodrahalli. A Large Self-Annotated Corpus for Sarcasm, 2018.
>
> [8] Yiyi Liu, Ruqing Zhang, Yixing Fan, Jiafeng Guo, and Xueqi Cheng. Prompt Tuning with Con-tradictory Intentions for Sarcasm Recognition, 2023.
>
> ---
>
>
> &#8595; &#8595; &#8595; **Continued below** &#8595; &#8595; &#8595;

---

> ### Author Response · Authors · 2023-11-20
> **Response to Reviewer K4wg (Part 4/4)**
>
> > In cases where the generated data is a combination of two data providers, such as books and wiki (a common situation), how can one determine which data source the generated content comes from?
>
> Thank you for your insightful comment and for pointing out this potential extension of our framework. Since our WASA framework is the first to achieve effective source attribution for LLM-generated data, we have focused on other important challenges in our paper, and hence did not consider the extension where the generated data is a combination of multiple data providers.
> However, such scenarios can in fact be naturally handled by our top-k source attribution (Sec. 4.1): **We can use the generated top-k watermarks to identify the k most likely data providers**, in order to account for cases where there are multiple data providers.
>
> To demonstrate our framework's capability in this context, we have crafted several case studies simulating examples of text that are combinations of two data providers. We select two pieces of text generated by different data providers and manually concatenate them. Subsequently, we use the concatenated text as the prompt for WASA-LLM to generate the top-3 watermarks. As an example, we have crafted the following text as the concatenation of the generated texts from two data providers, `gr-qc` (with watermark `U+200DU+2064U+200BU+200BU+200CU+200BU+200BU+200DU+2063U+200C`) and `quant-ph` (with watermark `U+2062U+2063U+200CU+2063U+2063U+2063U+200CU+200CU+200BU+200D`). In such cases, **our framework is able to produce the watermarks corresponding to both data providers among the top-3 generated watermarks**. Note that in the following examples, we manually visualize the watermarks for illustrative purposes, while in real cases, the watermarks remain invisible.
>
> `
> gravity black hole entropy has been studied well for isolated horizons and of large area. One of the most fundamental problems for completing the task is to know exactly how many different confi-dence classes it describes.
> \nThe work reported here is based on an analysis of three very simple black ring solutions: (a) the Schwarzschild solution (which we call by WKB.
> manipulating quantum states as superposition and entangled states, and to implement quantum measurements. Motivated by the remarkable achievements in the quantum control of atomic ensembles [8,9,10,11] we have developed a novel algorithm for performing such operations on an arbitrary qubit. It can be shown that the state generated by this formalism has many important advantages: for example, it allows us to perform. Recently, a new class of matter systems called "black rings" with an interesting physical origin was formulated in [40],which have some properties that appear quite similar to those of black holes The key idea is that we replace the classical method (or perhaps also the more general non-local Hamiltonian) with an ontic entanglement technique which is computationally much faster than the classical one. [WTM] U+200DU+2064U+200BU+200BU+200CU+200BU+200BU+200DU+2063U+200C[WTM] U+2062U+2063U+200CU+2063U+2063U+2063U+200CU+200CU+200BU+200D[WTM] U+2063U+200CU+200CU+200BU+200DU+2063U+2063U+200CU+200BU+2062
> `
>
> We have also performed other case studies, in which our framework is also able to generate the watermarks corresponding to both data providers among the top-3 generated watermarks. These results demonstrate the potential of our top-k source attribution to handle scenarios in which the generated data is a combination of multiple data providers.
> We have added these case studies in Appendix G.17 of our revised paper.
>
>
> ---
>
> Thank you again for your insightful comments. We hope that our clarifications and experimental results can improve your opinion of our paper.

---

> ### Comment · Reviewer_K4wg · 2023-11-21
> **Thanks for the detailed response!!**
>
> Thank you for the detailed response! The authors have effectively addressed my concerns, and I am inspired by the simple yet effective watermark techniques. As a result, I would like to increase my score.

---

> ### Author Response · Authors · 2023-11-22
> **Thanks to Reviewer K4wg**
>
> Dear Reviewer K4wg,
>
> Thank you so much for your positive feedback! We are deeply encouraged by your support and recognition of our work. Please kindly let us know if you have any further concerns regarding our response, which we are happy to address.

---

### Official Review · Reviewer_ViRU · 2023-11-06

**Soundness:** 3 good
**Presentation:** 4 excellent
**Contribution:** 2 fair
**Rating:** 6
**Confidence:** 3

**Summary:**

The authors tackle source attribution in large language models via a watermarking method. In particular, they introduce a simple watermark (a ten-token string of "invisible" unicode characters), together with a slight modification to the usual transformer architecture that allows the model to "switch" between generating standard text and watermark tokens. The authors then evaluate the proposed framework on datasets consisting of multiple data sources.

---

Score updated from 5 to 6 after rebuttal.

**Strengths:**

- The problem of source attribution is a critical problem in the context of deploying large language models in the real world. The authors directly tackle this problem and propose a simple framework to alleviate it.

 - The authors clearly list desiderata for a good watermarking framework, and argue how their proposed framework satisfies each of them. This makes the paper easy to follow.

- The authors consider a well-scoped setup (training GPT-2 on the ArXiv dataset) for their experiments.

- The authors do a great job of assessing the robustness of their watermarks.

**Weaknesses:**

# Evaluation
My main concern with this paper is with the choice of evaluation strategies.

First:
> To facilitate easy evaluations of the source attribution accuracy, for each data provider, we use the
sentences selected for watermarking (after removing the watermarks) as the inputs/prompts to the
trained WASA-LLM, and use the watermarks embedded into the generated texts for source attribution. This simplifies the evaluations because the corresponding data provider of the input sentence is
naturally the ground-truth source.

I am afraid this may be too close to "evaluating on the training set", and thus may significantly skew the reported results. In particular, at a minimum, I would like to see an evaluation on a "held-out" set of sentences from each data provider which were *not* selected for watermarking.

Second:
>  This simplifies the evaluations because the corresponding data provider of the input sentence is
naturally the ground-truth source. [...] Specifically, we first select 50 sentences from each data provider
[...] Next, we use the first 200 characters of every selected
sentence (without watermarks) as the input/prompt to the trained WASA-LLM which then generates
synthetic texts (by continuing the sentence) with a token length of 100. The watermark embedded
into the generated sentence is then used for source attribution, i.e., the source attribution is correct if
this watermark matches the watermark of the data provider corresponding to this sentence.

It is not directly clear to me that the generated text prompted by a particular sentence from the training set is "naturally" generated from the same source. For example, I could imagine prompting an LLM to answer a particular fact (e.g., contained in Wikipedia), but structure my query in a way that maximizes TF-IDF with a document from e.g. ArXiv. Then, "naturally", the model should be using the Wikipedia source, not the ArXiv one, when completing the prompt.


Minor concern:
- The proposed watermarking framework cannot be applied to already trained models, i.e., it requires a specific intervention during training. This may significantly limit its applicability.

**Questions:**

> our WASA-LLM almost never generates incorrect (unseen) watermarks
How often does that happen? In particular, is there a significant difference in the frequency of unseen watermarks when watermarking texts *not* in the training data? What about texts in the training data *not* selected for watermarking?

---

> ### Author Response · Authors · 2023-11-20
> **Response to Reviewer ViRU (Part 1/3)**
>
> We'd like to thank the reviewer for your insightful feedback.
>
> ---
>
> We would like to address your second question before the first one, because our response to the second question helps better support our response to your first question.
>
> > It is not directly clear to me that the generated text prompted by a particular sentence from the training set is "naturally" generated from the same source. For example, I could imagine prompting an LLM to answer a particular fact (e.g., contained in Wikipedia), but structure my query in a way that maximizes TF-IDF with a document from e.g. ArXiv. Then, "naturally", the model should be using the Wikipedia source, not the ArXiv one, when completing the prompt.
>
> Thank you for your insightful comment on how to determine the ground truth source for a generated sentence. We have added experiments to verify that **our method** of using the source of the prompt sentence as the ground truth source for the generated sentence **is indeed a reliable approach**, in addition to its benefit of simplifying the experimental evaluation.
>
> A natural and reliable method to find the ground truth source of a generated text is to consult the opinion of human experts. Therefore, we would like to show here that our method to determine the ground truth source is an **accurate approximation to human evaluations**. To avoid the substantial costs and resources associated with human evaluators (which is infeasible during the short rebuttal period), we have employed GPT-4, noted for its human-level performance across various benchmarks [1], as a surrogate 'human-like labeler'. Then, we examine whether the ground truth source determined by our method (i.e., using the source of the prompt sentence) aligns well with those determined by GPT-4.
> Specifically, we use GPT-4 to categorize generated texts into one of the ten ArXiv categories (i.e., data providers) using a carefully constructed prompt (shown below). After evaluating 500 generated texts, we have found that **89.6% of GPT-4's decisions align with our source determination method** (i.e., using the source of the prompt sentence). This validates that our method to determine the ground truth source of a generated text is a reasonable and reliable approach.
>
> We would like to add that employing GPT-4 as a 'human-like labeler' is only feasible in our controlled setting here because it requires including our prior knowledge about all sources and detailed descriptions of the sources (see the  prompt we used below). Moreover, it also incurs excessive costs in terms of monetary expenses and computations when the number of data providers is large. Therefore, we would like to clarify that this GPT-4-based method is not a realistic alternative method for source attribution, and is instead only employed here to verify the reliability of our method of source determination.
>
> We have added the experiments, results, and discussions here to Appendix G.10 of the revised paper. We will also discuss them in the main paper after revision to demonstrate the reliability of our method of source determination.
>
> ---
> The simplified prompt we have used in the experiment above is as follows (a detailed prompt is included in Appendix G.10 in our revised paper):
>
> ```
> Given below are 10 categories for texts from ArXiv papers with their descriptions. Please read the descriptions and classify the provided texts to one of the paper categories.
> The 10 categories are: hep-th, hep-ph, quant-ph, astro-ph, cs.CV, cs.LG, cond-mat.mes-hall, gr-qc, cond-mat.mtrl-sci, cond-mat.str-el.
> hep-th stands for High Energy Physics - Theory. This category includes research papers which are centered on theoretical concepts and mathematical models in high energy physics.
> ...
> cond-mat.str-el stands for Condensed Matter - Strongly Correlated Electrons. This category includes research papers focused on the study of solids and liquids in which interactions among electrons play a dominant role in determining the properties of the material.
> Note that you should only include the class in your reply and provide no explanations.
> Please classify the following sentence into one of the 10 categories, however, if you think that the sentence could be classified into multiple categories, you may give up to 3 most likely categories:
> ```
>
> REFERENCES
>
> [1] OpenAI. GPT-4 Technical Report. arXiv:2303.08774, 2023.
>
>
> ---
>
>
> &#8595; &#8595; &#8595; **Continued below** &#8595; &#8595; &#8595;

---

> ### Author Response · Authors · 2023-11-20
> **Response to Reviewer ViRU (Part 2/3)**
>
> > I am afraid this may be too close to "evaluating on the training set", and thus may significantly skew the reported results. In particular, at a minimum, I would like to see an evaluation on a "held-out" set of sentences from each data provider which were not selected for watermarking.
>
> Thank you for your insightful comment regarding our evaluation methodology. Below, we firstly provide the results for the additional experiments you have alluded to using the "held-out" sets as the prompt sentences, and then use further experiments to justify our choice of using the watermarked training data as the prompt sentences (in the original paper).
>
> Firstly, as you suggested, we have added 2 sets of experiments which use "held-out" sets of sentences as the prompt sentences: (1) "held out (unwatermarked training data)" in which we use the sentences that are used for training yet are not watermarked, and (2) "held out (validation data, not used for training)" in which we use sentences that are not used for training at all.
> The resulting source attribution accuracies are shown in the table below (averaged over five random seeds), which demonstrate that **our WASA framework still achieves decent source attribution accuracies when the prompt sentences are from the held-out sets**.
>
> | prompts | top-1 acc | top-3 acc | top-5 acc | Alignment of source with GPT-4 decisions
> | -------- | -------- | -------- | -------- | -------- |
> | watermarked training data (in original paper)  |  74.84% | 95.76% | 98.56% | 89.60% |
> | held out (unwatermarked training data)  |  66.16% | 90.60% | 95.28% | 81.60% |
> | held out (validation data, not used for training)  |  64.72% | 90.32% | 96.00% | 75.00% |
>
> Secondly, note that the reason why we have used watermarked training data as the prompt sentences in our evaluation was because it leads to simple and **reliable evaluations**. Here, we justify this using the experiment we adopted in our response above: We use GPT-4 to examine the reliability of the ground truth source determination when sentences from the held-out sets are used as the prompt sentences. The results (last column in the table above) show that when the prompt sentences are from unwatermarked training data (second row), **81.6%** of GPT-4's decisions align with the source of the prompt sentences; when the prompt sentences are from the validation data (third row), the alignment becomes **75.0%**. The results suggest that when the sentences from both held-out sets are used as the prompt sentences, our method to determine the ground truth source is still reasonably reliable. However, our ground truth source determination is the most reliable when sentences from watermarked training data are used as the prompt sentences, as we have done in our main experiments in the original paper. Therefore, the results justify the rationale behind our choice of using watermarked training data as prompts, because it **enhances the reliability of our source determination and hence the fidelity of our evaluation results**. We have also added the results and discussions here to Appendix G.10 of the revised paper.
>
>
> ---
>
>
> &#8595; &#8595; &#8595; **Continued below** &#8595; &#8595; &#8595;

---

> ### Author Response · Authors · 2023-11-20
> **Response to Reviewer ViRU (Part 3/3)**
>
> > The proposed watermarking framework cannot be applied to already trained models, i.e., it requires a specific intervention during training. This may significantly limit its applicability.
>
> You are correct that our framework applies intervention during training. However, as we have mentioned when discussing the **adaptability** property of our framework (e.g., see Sec. 2, point 6), our framework only applies mild modifications to the LLMs and hence **minor intervention** during LLM training. **This makes our framework applicable to a wide variety of LLMs** and hence does not limit its applicability.
>
> We would like to add that our framework is **the first to achieve effective source attribution for LLM-generated data**. Therefore, we leave it to future works to explore other source attribution frameworks that do not require intervention during training.
>
> ---
>
> > our WASA-LLM almost never generates incorrect (unseen) watermarks. How often does that happen? In particular, is there a significant difference in the frequency of unseen watermarks when watermarking texts not in the training data? What about texts in the training data not selected for watermarking?
>
> In our experiments, the number of incorrect (unseen) watermarks is **0**, which we have shown in Table 8 and discussed in the paragraph "Fine-grained Analysis of Source Attribution Errors" in Appendix F.1. As you suggested, we have added experiments using (1) the texts not in the training data or (2) the unwatermarked texts in the training data as the prompts. The results show that the number of incorrect (unseen) watermarks is still **0** in both cases. These results further corroborate the ability of our framework to learn an accurate mapping from the texts to watermarks. Thank you for the suggested experiments. We will revise the corresponding descriptions in the main paper to make it clearer.
>
> ---
>
> Thank you again for your constructive and insightful comments. We hope our clarifications and additional experiments can improve your evaluation of our paper.

---

> ### Author Response · Authors · 2023-11-22
> **Thanks to Reviewer ViRU**
>
> Dear Reviewer ViRU,
>
> Please allow us to thank you again for your valuable insights of our paper. We hope that our clarifications and additional experiments have addressed your concerns. We would be grateful if you could share any further feedback.

---

> > ### Comment · Reviewer_ViRU · 2023-11-22
> > **Response**
> >
> > I am satisfied with the authors' response and thus raise my score.

---

> ### Author Response · Authors · 2023-11-23
> **Thanks to Reviewer ViRU**
>
> Dear Reviewer ViRU,
>
> Thank you so much for your positive feedback! We are happy to hear that we have addressed your concerns. Your recognition of our work deeply encourages us.

---

### Official Review · Reviewer_aG26 · 2023-11-06

**Soundness:** 3 good
**Presentation:** 2 fair
**Contribution:** 2 fair
**Rating:** 5
**Confidence:** 2

**Summary:**

This paper addresses concerns surrounding the intellectual property of training data for large language models (LLMs) and proposes a watermarking solution for source attribution and data provenance. The authors introduce the WAtermarking for Source Attribution (WASA) framework, which satisfies key properties for effective source attribution, robustness, and adaptability to different LLMs. Empirical evaluations confirm that WASA effectively achieves source attribution and data provenance for LLM-generated synthetic texts.

**Strengths:**

- Interesting topic
- Nice summarization of desired watermark properties
- Extensive experiments

**Weaknesses:**

- Presentation could be improved
- Watermark can be easily removed
- Questionable threat model

**Questions:**

**Presentation could be improved**

The initial observation to make regarding this paper is that there is room for improvement in terms of its overall structure. Presently, it appears that the authors have included an extensive amount of information that they find interesting or significant for the readers. While this enthusiasm is commendable, it can make it challenging for readers to grasp the central idea at the outset.

Specifically, the methodology section poses a notable challenge as it lacks clarity in explaining how the method functions and what the underlying threat model is. This confusion arises from the authors blending their method with background information, and the threat model is informally introduced in the paper's introduction. It is important to emphasize that I am not suggesting a strict adherence to a standard layout, but rather, a reorganization of the paper's structure that would enhance the communication of the core concepts to the readers.

**Watermark can be easily removed**

Furthermore, there is a concern regarding the effectiveness of the watermark method proposed in the paper. The authors utilize invisible characters within sentences to maintain the imperceptibility of the watermark. However, it seems evident that a straightforward defense strategy would involve the removal of all invisible characters, rendering the watermark entirely ineffective.

**Questionable threat model**

Additionally, a related concern pertains to the use of these invisible characters. The authors appear to increase the vocabulary size to accommodate these rare characters. This raises the question of whether the insertion of a watermark requires manipulation of the tokenizer used by the target language model.

---

> ### Author Response · Authors · 2023-11-20
> **Response to Reviewer aG26 (Part 1/2)**
>
> We'd like to thank the reviewer for your valuable comments.
>
> ---
>
> > The initial observation to make regarding this paper is that there is room for improvement in terms of its overall structure. Presently, it appears that the authors have included an extensive amount of information that they find interesting or significant for the readers. While this enthusiasm is commendable, it can make it challenging for readers to grasp the central idea at the outset.
>
>
> Thank you for your constructive insights regarding the overall structure and clarity of our paper. Since we are the first work to achieve effective source attribution for LLM-generated data, we have indeed included an extensive amount of information, insights, and contributions into the paper as you have mentioned. Given the space constraints, we have prioritized the most significant contributions and results in the main paper and put additional details (e.g., examples, implementations, etc.) in the appendix. We will follow your suggestion to improve the structure of our paper to improve its readability.
>
> Here, we briefly discuss the **central ideas** of our paper for your convenience.
> Our WASA framework embeds unique and invisible Unicode characters into the training text data of each data provider as its watermark. Then, the framework operates in two distinct prediction spaces for text and watermark tokens to train the LLM. After the LLM is trained, the data generated by the LLM contains watermarks which allow for effective source attribution and data provenance. We have identified important properties that should be satisfied by a watermarking framework to achieve effective source attribution, and discussed how our WASA framework satisfies all these properties. We will follow your suggestion to revise the paper to make it easier to grasp these central ideas of our paper.
>
> ---
>
>
> > Specifically, the methodology section poses a notable challenge as it lacks clarity in explaining how the method functions and what the underlying threat model is. This confusion arises from the authors blending their method with background information, and the threat model is informally introduced in the paper's introduction. It is important to emphasize that I am not suggesting a strict adherence to a standard layout, but rather, a reorganization of the paper's structure that would enhance the communication of the core concepts to the readers.
>
> Regarding the blending of our method and background information, the subsection 'Preliminaries on LLMs' is where we introduce the essential background details on LLMs. We think these details such as the notations are important for understanding our method in the later sections. We will follow your suggestion to consider restructuring our methodology section (e.g., moving the background information to an earlier section) to further improve its clarity.
>
> About the threat model, we agree that a more formal discussion on it would make our paper more complete. We have focused on 6 essential properties of our WASA framework (Sec. 2). Therefore, due to space constraints, we could not expand on the threat model, which is only relevant to the robustness property among the 6 properties.
> Here, we provide more discussions on the threat model. We identify potential attackers as those intending to alter the LLM-generated text to remove IP acknowledgments to data contributors or alter input sentences to disrupt the watermark generation and hence the source attribution results. The attackers do not have access to the LLM itself, but they can query the model and are able to modify the generated outputs. The attackers may also possess tools that can remove the Unicode characters (hence the watermark) inside a text. As a result, the attacks could involve tampering with the watermark or the synthetic text itself. Our regeneration defense mechanism, detailed in Table 2, is able to effectively defend against these attacks. We will also revise our paper to include a more explicit description of the threat model.
>
> ---
>
>
> &#8595; &#8595; &#8595; **Continued below** &#8595; &#8595; &#8595;

---

> ### Author Response · Authors · 2023-11-20
> **Response to Reviewer aG26 (Part 2/2)**
>
> > Furthermore, there is a concern regarding the effectiveness of the watermark method proposed in the paper. The authors utilize invisible characters within sentences to maintain the imperceptibility of the watermark. However, it seems evident that a straightforward defense strategy would involve the removal of all invisible characters, rendering the watermark entirely ineffective.
>
> Thank you for this natural and insightful question. We would like to clarify that **removing all invisible characters in fact does not render our watermark ineffective**, which we explain below from 3 perspectives.
>
> 1. Watermark Regeneration Strategy
>
> To demonstrate the capability of our method in defending against the removal of all invisible characters (the removal of watermarks), we have introduced a strategy of watermark regeneration, which is thoroughly discussed in Sec. 4.2 of our paper. As highlighted, the source attribution accuracy (top-3 accuracy) of this regeneration defense is **71.60% (93.76%)** which is comparable to the original **74.84% (95.76%)**. Thus, **our watermark regeneration is an effective defense mechanism** to address the straightforward removal of watermarks.
>
> 2. Impractical to Remove All Invisible Characters
>
> We would like to clarify that the removal of all invisible characters in the text is not a practical strategy in some scenarios, because the original unwatermarked text could include informative invisible characters. For instance, there can be data originally containing invisible Unicode characters within the training data, such as the CC-News dataset [1]. It comprises about 700K English-language news articles from various global news websites. The invisible Unicode characters in this dataset are essential for websites for design purposes. Therefore, removing all Unicode characters can adversely affect the textual presentation and integrity.
>
> 3. Removing Watermarks Has No Impact in Some Scenarios
>
> In many scenarios, source attribution can be performed immediately after the text is generated by an LLM. For example, when our WASA framework is used to develop a chatbot for navigating numerous digital resources in a public library, we can immediately find the source of the generated text to examine the credibility of the text. In these scenarios, source attribution has already been completed immediately after the text generation, and therefore subsequent removals of the Unicode characters (hence watermarks) do not affect the source attribution at all.
>
>
>
> References:
>
> [1] Felix Hamborg, Norman Meuschke, Corinna Breitinger, and Bela Gipp. news-please: A Generic News Crawler and Extractor. In Proceedings of the 15th International Symposium of Information Science, pp. 218-223, 2017.
>
> ---
>
>
> > Additionally, a related concern pertains to the use of these invisible characters. The authors appear to increase the vocabulary size to accommodate these rare characters. This raises the question of whether the insertion of a watermark requires manipulation of the tokenizer used by the target language model.
>
> Your understanding is correct. Our framework indeed involves a **minor manipulation** of the original language model tokenizer to increase the vocabulary size. Specifically, the original vocabulary size of the GPT2 tokenizer is 50257 (0 to 50256), and we have added 6 watermark tokens (50257 to 50262) to the vocabulary. Similarly, the tokenizer of OPT has a vocabulary size of 50272 (0 to 50271), and our 6 added watermark tokens range from 50272 to 50277. Additionally, as discussed in Sec. 3.3, we have introduced another special token [WTM] to guide the generation of watermarks. Note that these modifications are minimal, representing an approximate 0.01% increase in the original vocabulary size. Therefore, these minimal changes can be integrated with existing tokenizer, and hence align with our property of **adaptability** (Sec. 2).
>
> ---
>
> Thank you again for your feedback. We hope that our clarifications can improve your opinion of our paper.

---

> ### Author Response · Authors · 2023-11-22
> **Thanks to Reviewer aG26**
>
> Dear Reviewer aG26,
>
> We would like to thank you again for the time and effort you have dedicated to reviewing our paper. Please kindly let us know if there are any further queries regarding our response. Your further feedback will be greatly appreciated.

---

> > ### Comment · Reviewer_aG26 · 2023-12-01
> >
> > Thank you for the clarification!
> >
> > > Your understanding is correct. Our framework indeed involves a **minor manipulation** of the original language model tokenizer
> >
> > I am confused about this point since I am not sure what the threat model is. Let's refer to the identity attempting to insert watermarks as the attacker. Is the attacker responsible for contributing training data, or is the attacker responsible for model training, or does the attacker control the entire process, including training and data collection? To me, contributing training data seems the most reasonable, but considering that the attacker could manipulate the tokenizer, it appears that the attacker is the one controlling the model. I believe it would be better if the authors could provide clarification on this matter.

---

### Meta-Review · Area_Chair_aZd9 · 2023-12-06

**Metareview:**

**Summary**

The paper proposes a watermarking framework called WASA (Watermarking for Source Attribution) to address concerns about source attribution and data provenance in large language models (LLMs). By embedding watermarks in generated texts, WASA aims to identify the data sources responsible for generating those texts and verify whether data from specific providers has been used to train the LLM. The framework aims to satisfy key properties such as accuracy, robustness, adaptability, and scalability. Empirical evaluations show that WASA effectively achieves source attribution and data provenance.

**Strengths**

* Addresses an important issue of source attribution and data provenance in LLMs.
* Proposes a novel watermarking solution using imperceptible Unicode characters.
* Presents a comprehensive framework that addresses multiple desirable properties.
* Provides extensive empirical evaluations to support the framework's effectiveness.

**Weaknesses**


* **Presentation:** Some reviewers found the presentation to be difficult to follow and suggested simplifying certain statements and providing more examples.
* **Scalability:** The framework's accuracy decreases as the number of data providers increases, raising concerns about scalability.
* **Experimental diversity:** Experiments are limited to certain types of datasets, such as academic papers and books, casting doubts on the framework's versatility. One reviewer raised the concerns on the generalization of the approach for the noisy and practical datasets.
* **Limited applicability:** The watermarking framework cannot be applied to pre-trained models and requires intervention during training, which may limit its applicability.

**Justification For Why Not Higher Score:**

Reviewers highlight areas for improvement and suggest further research to address scalability concerns, the robustness and generalizability of the approach to less curated and noisy datasets, and limited applicability for not being able to be applied to pre-trained models.

**Justification For Why Not Lower Score:**

N/A

---

### Decision · Program_Chairs · 2024-01-16

Reject